# Hybrid soliton dynamics in liquid-core fibres

Mario Chemnitz [1], Martin Gebhardt[2,3], Christian Gaida[2], Fabian Stutzki[2], Jens Kobelke[1], Jens Limpert[2,3,4], Andreas Tünnermann[2,3,4] & Markus A. Schmidt [1,5]

The discovery of optical solitons being understood as temporally and spectrally stationary optical states has enabled numerous innovations among which, most notably, super-continuum light sources have become widely used in both fundamental and applied sciences. Here, we report on experimental evidence for dynamics of hybrid solitons—a new type of solitary wave, which emerges as a result of a strong non-instantaneous nonlinear response in $CS_2$-filled liquid-core optical fibres. Octave-spanning supercontinua in the mid-infrared region are observed when pumping the hybrid waveguide with a 460 fs laser (1.95 μm) in the anomalous dispersion regime at nanojoule-level pulse energies. A detailed numerical analysis well correlated with the experiment uncovers clear indicators of emerging hybrid solitons, revealing their impact on the bandwidth, onset energy and noise characteristics of the supercontinua. Our study highlights liquid-core fibres as a promising platform for fundamental optics and applications towards novel coherent and reconfigurable light sources.

[1] Leibniz Institute of Photonic Technology, Albert-Einstein-Strasse 9, Jena 07745, Germany. [2] Institute of Applied Physics, Abbe Center of Photonics, Friedrich-Schiller-University Jena, Albert-Einstein-Strasse 15, Jena 07745, Germany. [3] Helmholtz-Institute Jena, Froebelstieg 3, Jena 07743, Germany. [4] Fraunhofer Institute for Applied Optics and Precision Engineering, Albert-Einstein-Strasse 7, Jena 07745, Germany. [5] Otto-Schott-Institute of Material Research, Friedrich-Schiller-University of Jena, Fraunhoferstrasse 6, Jena 07743, Germany. Correspondence and requests for materials should be addressed to M.C. (email: mario.chemnitz@leibniz-ipht.de)

Optical solitons as temporally and spectrally localised optical entities have always been one of the most fascinating outcomes of nonlinear light–matter interactions in optical waveguides. Their discovery opened up new perspectives in telecommunications[1], laser development[2] and fundamental science such as the modelling of relativistic systems[3] or statistical rogue wave formation[4]. Most notably, the break-up of intense ultrafast pump pulses into a multitude of solitons distributed over several spectral octaves pushed the development of broadband supercontinuum sources—a technology that is indispensable in several applications such as optical coherence tomography[5], metrology[6] and spectroscopy[7].

The limitations of commercial supercontinuum systems go along with the constraints of silica fibres in terms of transmission bandwidth and nonlinearity $(n_2 = 3.2 \times 10^{-16}\,\mathrm{cm^2\,W^{-1}})$[8]. To overcome these limitations, current research on soliton-based supercontinuum generation is driven by novel and hybrid material waveguides. Well-known concepts from silica fibres, such as dispersive wave generation[9, 10] and soliton-self frequency shifts[11], have been transferred to soft-glass systems (e.g., chalcogenides, tellurides, fluorides), taking advantage of their significantly higher nonlinearity and wider transmission windows, particularly towards mid-infrared wavelengths. Furthermore, ultraviolet wavelengths have been accessed via soliton interactions in noble-gas-filled hollow-core fibres[12, 13], bridging the wavelength gap between visible and extreme ultraviolet light sources[14]. These fibres possess much potential not only from the applications' perspective, but also for the study of new physics. For example, harnessing the strong Raman response of bi-atomic gases has revealed soliton pairing as one of the core features in the spectral output[15].

Likewise, liquid-core optical fibres (LiCOF) promise the efficient exploitation of another nonlinear response with picosecond-long decay times, which is unique for liquids. This highly non-instantaneous response originates from the strong reorientation (induced anisotropy) and libration of elongated liquid molecules in an optical field. The resulting nonlinear refractive index can be comparable to those of soft-glass systems, being up to two orders of magnitude higher than that of silica (e.g., nitrobenzene with $n_2 = 670 \times 10^{-16}\,\mathrm{cm^2\,W^{-1}}$ [8] and carbon disulfide ($CS_2$) with $n_2 \leq 285 \times 10^{-16}\,\mathrm{cm^2\,W^{-1}}$ [16]). However, it is questionable whether this kind of nonlinearity can be harvested for supercontinuum generation and, more specifically, how it affects soliton formation and fission.

In an initial theoretical work in 2010, Conti et al.[17] investigated the impact of a long-lasting nonlinear response on pulse propagation. They found that pulses that are substantially shorter than the response time of the liquid (pulse width $\tau_0 \ll \tau_R = \int t R(t)\mathrm{d}t$) experience a potential not defined by their own intensity, as in the case of classical solitons, but by the material response $R(t)$ due to an optical memory effect. For this case, they deduced a quasi-linear propagation equation that yields a new class of solitary waves, termed non-instantaneous solitons, which appear like modes in a linear time potential. Similar to classical solitons, those states require anomalous dispersion, i.e., that the second derivative of the propagation constant $\beta_2 = \partial_\omega^2 \beta|_{\omega_0}$ ought to be negative. Those states exhibit novel features such as significantly reduced sensitivity to phase noise, which promises a new family of highly coherent supercontinuum sources not suffering from noise-dictated spectral bandwidth fluctuations.

The theory presented by Conti et al. predicts those states in LiCOFs, but it remained unclear how this work connects to previous studies of soliton fission and supercontinuum generation in those fibres[18, 19]. Furthermore, their model neglects the instantaneous response from the electrons and the rise time of the non-instantaneous nonlinearity, which both occur in the realistic response functions of liquids[16, 20]. Hence, the determination of the properties of those states for a realistic material response is a matter of conducting more complete numerical studies that clearly correlate with the experimental findings.

The experimental evidence for the existence of non-instantaneous solitons is hard to find since most liquids are normal dispersive up to mid-infrared wavelengths, where high loss dominates. The first attempts to obtain supercontinuum generation in the anomalous dispersion regime of liquids involved water-filled hollow-core photonic crystal fibres[21, 22] or photonic crystal fibres selectively filled with $CCl_4$[23]. However, the large absorption of water in the near-infrared or the low contribution of the non-instantaneous molecular nonlinearity in $CCl_4$[20] has prevented the observation of new soliton dynamics so far. In fact, only a few hydrogen-free liquids, like $CS_2$, offer sufficient transparency together with a strong non-instantaneous nonlinear contribution, up to 90% for picosecond pulses[16]. Yet, guidance along several metres in $CS_2$ remains challenging because of attenuation, preventing a direct investigation of fundamental solitary states.

In this work, we reveal the potential of using supercontinuum spectra generated in $CS_2$-core fibres to obtain first insights into non-instantaneous soliton dynamics. We theoretically investigate in detail the impact of a highly non-instantaneous nonlinearity including the realistic response of $CS_2$, first, on soliton formation and, second, on fission-based supercontinuum generation with the focus on bandwidth and coherence. We access the anomalous dispersion regime of liquid-core fibres using step-index geometries and a thulium fibre laser, and demonstrate efficient soliton-driven supercontinuum generation in a $CS_2$/silica fibre pumped with sub-picosecond pulses at 1.95 μm centre wavelength. Simulations, including both general and specialised models, unambiguously reveal spectral observables as measurable indicators of a hybrid soliton dynamic.

## Results

**Hybrid soliton formation.** Typical ultrashort pulses with a duration of tens to hundreds of femtoseconds experience each nonlinearity in conventional glasses (quasi) instantaneously, as they have their origin mainly in sub-femtosecond electronic excitations (Kerr effect) and partially in stimulated Raman scattering processes with response times of a few tens of femtoseconds. Elongated liquid molecules such as $CS_2$ instead possess dominant non-instantaneous nonlinearities in case of ultrashort pulses that are significantly shorter than the nonlinear response time of the liquid (e.g., $CS_2$ with 1.6 ps). In the following, we discuss the impact of such a response on the propagation of sub-picosecond pulses for $CS_2$-based LiCOFs.

The nonlinear response of liquid $CS_2$ is inherently hybrid for the pulse width considered here and the instantaneous contribution from the electronic motions cannot be neglected. Therefore, we define a molecular fraction factor $f_\mathrm{m} = n_{2,\mathrm{mol}}/(n_{2,\mathrm{el}} + n_{2,\mathrm{mol}})$ to quantify the weight between molecular ($n_{2,\mathrm{mol}}$) and electronic ($n_{2,\mathrm{el}}$) nonlinearities. Notably, the total nonlinear refractive index of $CS_2$ ($n_{2,CS_2} = n_{2,\mathrm{el}} + n_{2,\mathrm{mol}}$), and thus the molecular fraction, depends on the input pulse duration (see Methods)[16]. Throughout this work, we use 450 fs pulses (full width at half maximum $\tau_\mathrm{FWHM} = 2\ln(1+\sqrt{2})\tau_0$), which result in 85% molecular and 15% electronic contribution to the total nonlinearity of the system (i.e., $f_\mathrm{m} = 0.85$).

We theoretically compare two hypothetical cases with the realistic hybrid $CS_2$ system to separate the influences of instantaneous and non-instantaneous contributions on nonlinear pulse propagation. All cases assume identical fibre parameters (dispersion, nonlinear index, loss; see Methods), but differ in the

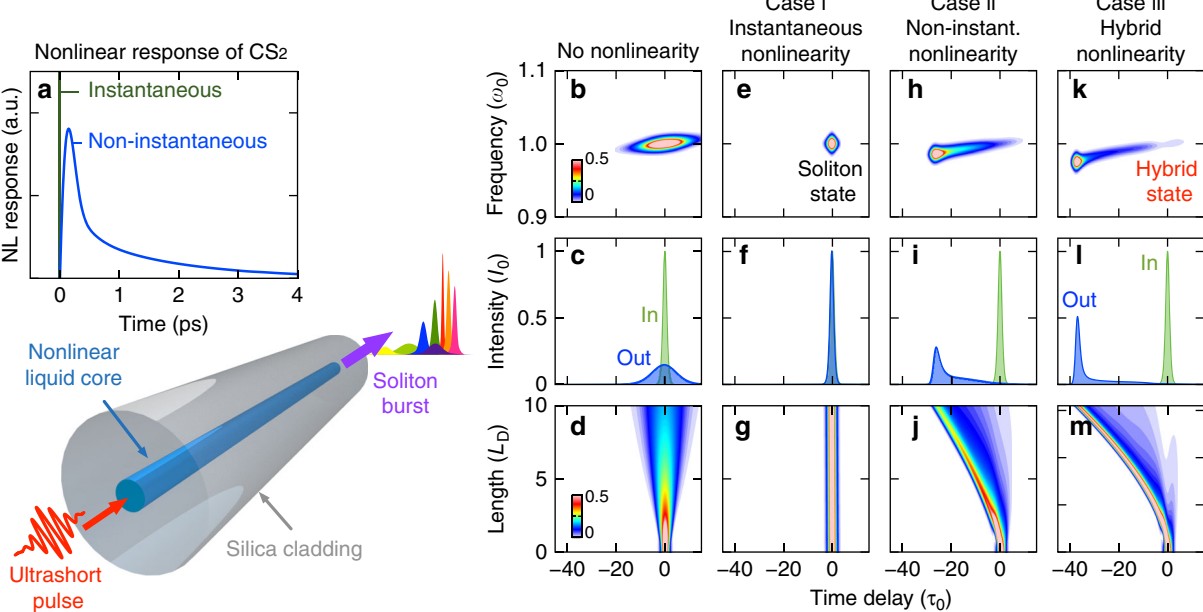

**Fig. 1** Characteristics of hybrid solitons. **a** Principle of soliton fission in a hybrid liquid/silica fibre possessing a typical highly non-instantaneous nonlinearity (here $CS_2$ according to[16]). **b**–**m** Pulse propagation characteristics along 10 dispersive lengths pumped with a sech-squared pulse ($\tau_{\text{FWHM}} = 450$ fs) in the anomalous dispersion regime (const. $\beta_2$, const. $\gamma_0$) for four different lossless systems: **b**–**d** no nonlinearity, **e**–**g** entirely instantaneous nonlinearity with soliton number $N = 1$, **h**–**j** entirely non-instantaneous nonlinearity based on $CS_2$ and **k**–**m** hybrid nonlinearity with both instantaneous (15%) and non-instantaneous contributions (85%). The *linear colour scale* represents peak power normalised to 50% of the input peak power

temporal responses. The first hypothetical system (case i) possesses only an instantaneous response (green line in Fig. 1a), the second system (case ii) has an entirely non-instantaneous response (blue line in Fig. 1a) and the third system (case iii) contains the realistic $CS_2$ response function (15% instantaneous and 85% non-instantaneous).

In an isotropic instantaneous system (case i), the group velocity dispersion (GVD) can be compensated by accumulating nonlinear phases (Kerr effect) during propagation. In the case of ideal phase compensation in a lossless medium, a fundamental classical soliton is formed, which is characterised by the soliton number $N = \sqrt{\gamma_0 P_0 \tau_0^2 |\beta_2|^{-1}} = 1$, where $\gamma_0$ is the nonlinear parameter of the waveguide mode and $P_0$ is the initial peak power. The soliton preserves its shape in time (Fig. 1f, g) and spectrum during propagation, visualised by a localization in the spectrogram in Fig. 1e at 10 dispersive lengths $L_D = \tau_0^2 |\beta_2|^{-1}$. This is fundamentally different to linear pulse propagation, where the pulse undergoes dispersive broadening and the peak power drops strongly, by 85% (Fig. 1b–d).

If we now consider an entirely non-instantaneous nonlinear medium (case ii), an asymmetric pulse shape emerges during the propagation along 10 $L_D$ (Fig. 1i, j). The pulse shows a comet-like characteristic in the spectrotemporal domain (Fig. 1h). The trailing confinement of the pulse mainly maintains its pulse width and bandwidth after an initial formation process. The pulse characteristics are very similar to those of non-instantaneous solitons, or 'linear' states, described by Conti et al.[17], where, however, they assumed an ideal exponential response function. This simulation shows that states with common temporal features, like the dispersive front, exist in realistically shaped response functions, too. According to the theory, the dispersive pulse characteristic results from the relatively low ratio of the response time and pulse width (here $\tau_R/\tau_0 = 5$), which is hard to improve practically in this particular system.

Finally, we consider the hybrid nonlinear response containing both electronic (fast) and molecular (slow) contributions

(case iii). We find an intermediate state with spectrotemporal confinement in between the instantaneous (case i) and the non-instantaneous (case ii) systems. It still features a dispersive front (Fig. 1l, m), but a temporally shorter trailing localization after 10 $L_D$ compared to case ii. The peak power drops along the last five dispersive lengths by only 10%. With essential features from both nonlinear contributions, we refer to this mixed state as a hybrid solitary wave (HSW).

**Fission of hybrid solitons**. We continue to investigate the influence of the non-instantaneous response on the fission of HSWs and on their spectral features in supercontinua. Soliton fission appears at high peak powers in classical Kerr systems as a result of perturbations on the soliton propagation by third-order dispersion or Raman scattering. As a consequence, higher-order solitons ($N > 1$) fall apart into a series of fundamental solitons ($N = 1$) each at different wavelengths and each potentially generating a dispersive wave on the normal dispersion side[24]. The total bandwidth of this complex process depends strongly on the fibre dispersion and its nonlinear gain, but also on the temporal characteristics of the nonlinear response of the core material.

We again compare the evolutions of intense sech-squared pulses ($N = 92$) in our two hypothetical systems (case i and ii) with the realistic $CS_2$ response (case iii) to elucidate the impact of the different nonlinear contributions. Here, we consider lossy systems with higher-order dispersion and wavelength-dependent nonlinearity (see Methods) to get as close as possible to realistic systems. The process is simulated by numerically solving the generalised nonlinear Schrödinger equation (GNLSE)[24] for the fundamental mode of the fibre including both instantaneous and non-instantaneous nonlinearities weighted by the molecular fraction $f_m$ (see Methods). Note that ideal solitary solutions do not exist in lossy systems. However, we continue naming observed solitary waves 'solitons' for simplicity without implying the mathematical integrability of the system.

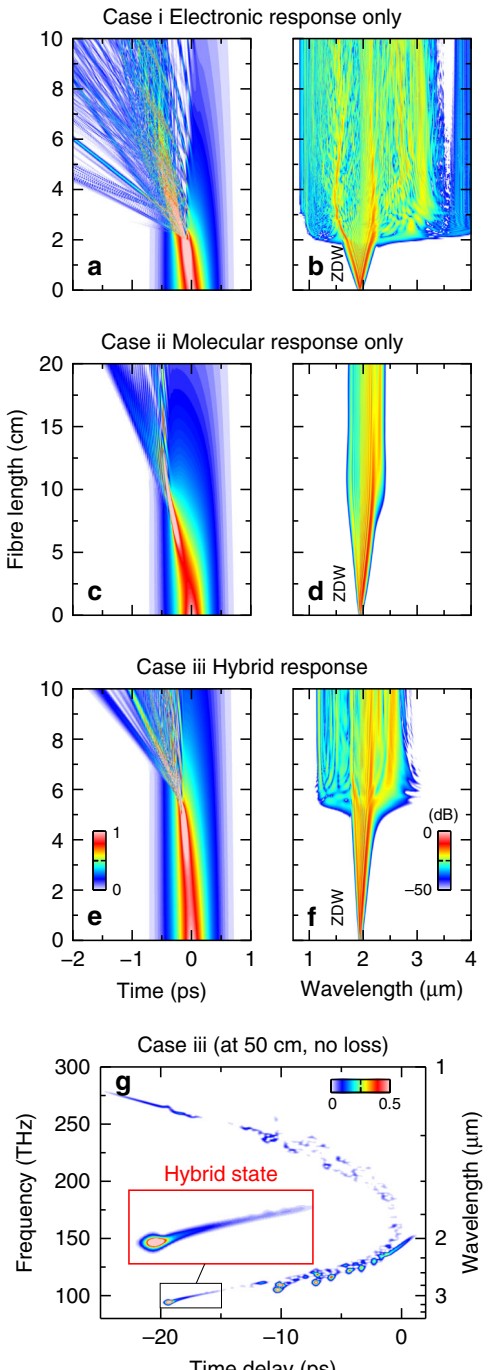

**Fig. 2** Impact of hybrid soliton dynamics on supercontinuum generation. Comparison of **a**, **c**, **e** temporal and **b**, **d**, **f** spectral evolutions of a high-power pulse (10 kW peak power, $\tau_{\text{FWHM}} = 450$ fs, $\lambda_p = 1.95$ m) in three nonlinear systems with the same loss, dispersion and nonlinear parameters, but different nonlinear responses: **a**, **b** entirely instantaneous response ($f_m = 0$), **c**, **d** entirely non-instantaneous response ($f_m = 1$) and **e**, **f** hybrid response of CS$_2$ ($f_m = 0.85$). The linear *colour scale* in the temporal domain **a**, **c**, **e** represents power normalised to the peak power of the initial pulse; the power scale in the spectral domain **b**, **d**, **f** is logarithmic (−50 to 0 dB). ZDW stands for zero-dispersion wavelength. **g** Spectrogram of the hybrid system assuming lossless propagation over 50 cm. The *linear colour scale* represents power normalised to 50% of the initial peak power

The instantaneous system (case i) shows conventional soliton fission: after initial self-phase modulation (SPM), a burst of fundamental solitons is released at a fission length of ~2 cm (Fig. 2a, b). Here, the strongly compressed pulse breaks up into multiple fundamental solitons on the long wavelength side, which shed energy towards shorter wavelengths via the generation of dispersive waves (i.e., Cherenkov radiation)[24], leading overall to a bandwidth that is just limited by absorption. The non-instantaneous system (case ii), however, shows the characteristic dynamics for linear states: initially, temporally confined wave packets are formed after a much longer fission length of 10 cm. After the initial compression, only a few well-separated linear states are formed (only the first one is visible in Fig. 2c). Their temporal characteristics reveal confined trailing features and dispersive fronts similar to the non-instantaneous states in Fig. 1j. In the spectral domain (Fig. 2d), we observe a significantly reduced bandwidth compared with the instantaneous system (case i).

The hybrid system (case iii) describes an intermediate situation in terms of bandwidth and fission length, with the fission point at 4 cm (Fig. 2e, f). The reduced spectral bandwidth and the delayed supercontinuum onset point in Fig. 2f clearly indicate a modification of classical solitary waves by the non-instantaneous response. Moreover, the spectrotemporal visualisation of a simulation without loss after 0.5 m propagation in Fig. 2g reveals the comet-like characteristics of HSWs (Fig. 1k). Conclusively, the spectral and temporal dynamics of solitary waves change drastically in presence of the non-instantaneous nonlinearity. In such systems, the classical soliton number $N$ can only serve as an upper estimate for the number of solitary waves created in the hybrid system because of the new dynamics and fibre losses.

**Coherence of hybrid soliton-based supercontinua.** A key characteristic of fission-based spectra is their reproducibility, i.e., the pulse-to-pulse coherence. The temporal jitter and spectral positions of the fundamental solitons at the fission point highly depend on the shape, phase and peak power of the initial pulse. Conventional soliton-driven supercontinuum sources lack pulse-to-pulse coherence due to the strong impact of input noise at large soliton numbers ($N > 10$), i.e., at high peak powers or long pulse durations (>200 fs)[24]. This puts high demands on both pump laser and fibre designs for applications where a high degree of coherence is necessary, e.g., pulse recompression[25, 26] or frequency comb metrology[27, 28].

Here, we compare the pulse-to-pulse coherence of the hybrid system (case iii) with the classical instantaneous system (case i) by solving the GNLSE for 100 individual pulses with identical initial parameters (pulse energy $\mathcal{E}_p = 1$ nJ, $\tau_{\text{FWHM}} = 450$ fs, $\lambda_p = 1.95$ μm, $N = 41$) but white photon noise on the input field as the simplest approach (one photon per frequency channel with random phase)[24]. The impact of phase noise on pulse-to-pulse spectral stability is quantified using the definition of the first-order degree of coherence[24]:

$$\left| g^{(1)}_{mn}(\lambda) \right| = \left| \frac{\left\langle E^*_m(\lambda) E_n(\lambda) \right\rangle}{\sqrt{\left\langle |E_m(\lambda)|^2 \right\rangle \left\langle |E_n(\lambda)|^2 \right\rangle}} \right|, \tag{1}$$

where $E(\lambda)$ denotes the electric fields at wavelength $\lambda$, $m$ and $n$ denote the indices of the individual spectra ($m \neq n$) and the angular brackets refer to an ensemble average.

The individual output spectra of the instantaneous system (Fig. 3b) show strong intensity fluctuations on the order of 30 dB across the entire bandwidth. Even small amounts of photon noise dictate the nonlinear phase conditions and, thus, the spectral

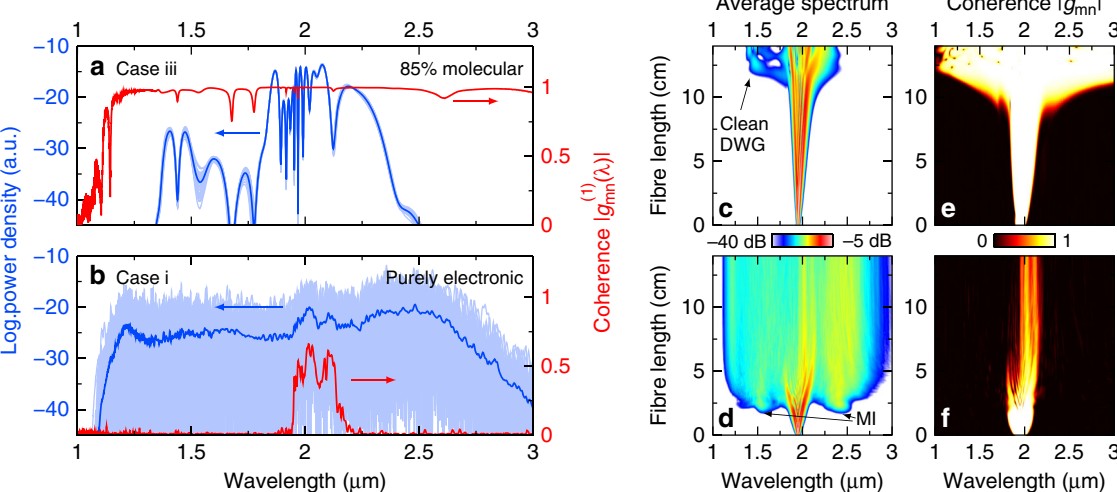

**Fig. 3** Impact of hybrid soliton dynamics on coherence. Ensemble and coherence analysis of the CS$_2$/silica LiCOF output (core diameter: 4.7 µm, length: 14 cm) for (**a**) the hybrid CS$_2$ response ($f_m = 0.85$) and (**b**) an entirely instantaneous response ($f_m = 0$). The *light blue lines* refer to 50 individual spectra calculated under identical input conditions ($\mathcal{E}_p = 1$ nJ, $\tau_{FWHM} = 450$ fs, $\lambda_p = 1.95$ µm, $N = 41$) with photon noise, the *solid blue lines* to the corresponding averages. The *red lines* represent the first-order coherence (right-handed axes). **c**, **d**. Evolution of the average spectrum along the fibre for each system (log. *colour scale* indicating spectral power: −40 to −5 dB). DWG: dispersive wave generation, MI: modulation instabilities. **e**, **f** Evolution of the first-order coherence for each system (*linear colour scale*: 0 to 1)

locations of solitons and associated dispersive waves. This high susceptibility to noise removes all fine features from the averaged output spectrum, leading to a flat spectral shape (blue line in Fig. 3b). In particular, the spectral lobes next to the pump at 1.6 and 2.4 µm in Fig. 3d rising at the fission length of 2 cm reveal modulation instabilities as the dominant broadening effect. Modulation instabilities are well known to occur at large soliton numbers ($N > 10$) as a result of noise-dictated chaotic soliton fission[29]. Accordingly, the coherence between subsequent broadband spectra vanishes beyond the fission point (Fig. 3b, f).

The hybrid system, however, is remarkably less susceptible to initial noise even at such long pulse widths. Variations between individual spectra are hardly visible in Fig. 3a (light blue lines) and distinct spectral features can still be observed in the average spectrum near 1.4 and 2.2 µm (blue line in Fig. 3a). Furthermore, the calculated spectra reveal a clear soliton fission process with the definite emergence of a soliton and a dispersive wave, which is in sharp contrast to the symmetric spectral side lobes in the case of modulation instabilities. This pulse-to-pulse spectral stability correlates with a perfect first-order coherence across the entire bandwidth (Fig. 3a, e), which clearly distinguishes hybrid systems from instantaneous systems.

The lesser susceptibility to noise can be explained by the comparably slow molecular response of CS$_2$, which 'stiffens' the nonlinear phase against fast temporal fluctuations. Initial fluctuations average out because of the convolution of the optical pulse with the slow material response that acts as optical phase rectification. Also, the later appearance of the fission point in the hybrid system at 12 cm (Fig. 3c) can be understood as a result of the reduced impact of noise, or strongly inhibited modulation instabilities. Note that the high level of coherence critically depends on the actual value of the molecular fraction. For the system parameters chosen in this work, only 5% less molecular contribution (i.e., $f_m = 0.8$) reduces the first-order coherence significantly (see Supplementary Fig. 4).

In conclusion, the dynamics of HSWs imprint significant signatures onto the generated spectra that are fundamentally different to those of classical supercontinuum systems. These signatures are a reduced bandwidth, higher supercontinuum

onset energy accordingly to the longer fission length, and a higher pulse-to-pulse coherence, with the latter being indicated by distinct spectral features and a clean fission process in the recorded average spectra. Thus, measurements of the spectral fingerprint (i .e., spectra over input power) of such a hybrid system will unambiguously reveal the impact of a long-lasting, non-instantaneous response on the soliton dynamics, similar to soliton-self frequency shifts that can be attributed to Raman effects in conventional systems[24].

Also note that the chosen pulse width provides a fair balance for making HSWs visible in the spectra. The pulses are sufficiently long to obtain a large molecular fraction of 85%, but short enough to ensure an acceptable ratio between the response time and pulse width $\tau_R/\tau_0$, fulfilling the fundamental assumptions for non-instantaneous solutions given by Conti et al., i.e., the Conti condition. Although shorter pulses would increase the ratio $\tau_R/\tau_0$, matching the Conti condition better, it implies smaller molecular fractions, which makes the identification of the spectral signatures more challenging (see Supplementary Fig. 3).

**Liquid-core fibres as platform for supercontinuum generation.** Soliton-driven light generation requires operation in the anomalous dispersion domain, which is hard to access in liquids with common ultrafast laser sources since the material dispersion of most liquids is normal far beyond 2 µm. Hence, spectral broadening in LiCOFs has mainly been demonstrated in the normal dispersion regime so far. Large Raman gains were shown in CS$_2$-core step-index fibres[30, 31], and supercontinuum generation based on SPM was achieved, featuring spectral bandwidths of up to 1.2 µm (127 THz)[32, 33]. However, soliton-based light generation has not been directly targeted in CS$_2$-LiCOFs, yet.

The large refractive index of CS$_2$ $n_{CS2} = 1.582 > n_{SiO2} = 1.438$ at $\lambda = 2$ µm) enables light guidance in straightforward-to-fill silica capillaries based on total internal reflection (Fig. 4b). A detailed analysis of the dispersion of such CS$_2$/silica step-index fibres reveal an anomalous dispersion domain below 2 µm, opening the soliton domain for state-of-the-art thulium fibre lasers (Fig. 4a).

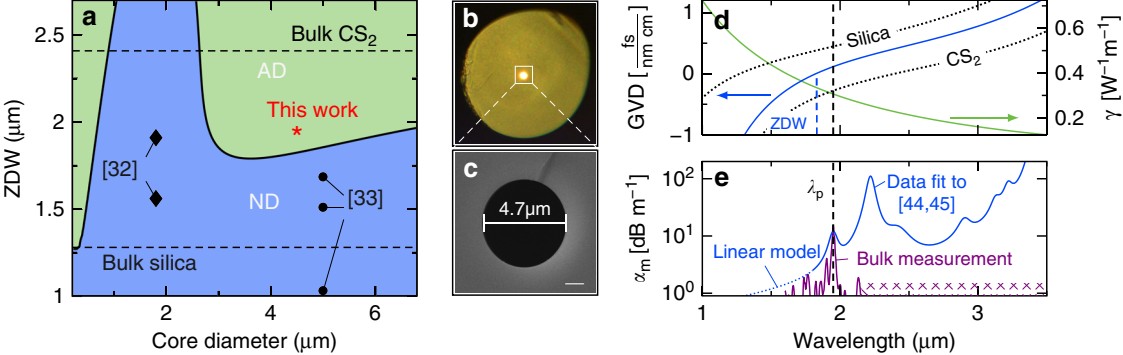

**Fig. 4** Optical properties of CS$_2$/silica step-index fibres. **a** GVD map: the spectral position of the ZDW for various core diameters (at 20 °C, 1 atm). *AD* anomalous dispersion, *ND* normal dispersion. The *black marks* indicate the work by other groups reported earlier. The *red star* refers to the experiments reported here. **b** Transmission microscope picture of a filled liquid-core step-index fibre and **c** electron microscope image of the capillaries used in this work. **d** GVD and nonlinear coefficient of the fundamental mode (HE$_{11}$) of the CS$_2$/silica fibre. **e** Estimated modal attenuation coefficient $\alpha_m$ compared to 1 m bulk CS$_2$ absorption measured up to 2.15 μm. The *vertical dashed line* indicates the pump wavelength $\lambda_p$

The strong waveguide dispersion of the hybrid fibre shifts the zero-dispersion wavelength (ZDW), which separates the normal from the anomalous dispersion domain, far below the zero dispersion of bulk CS$_2$ and the thulium laser line. It is a remarkable feature of the hybrid CS$_2$/silica system that anomalous dispersion can be achieved for core diameters (3–6 m) larger than the operation wavelength, which is in sharp contrast to soft-glass/silica fibres where subwavelength core diameters are required[9]. This enables high coupling efficiencies to the fundamental mode (HE$_{11}$), as well as robust guidance (i.e., sufficiently large V-parameters) even at wavelengths beyond the pump wavelength (see Supplementary Fig. 5). Such favourable conditions are not possible in the common domain around the first minimum of the ZDW at submicrometre core diameters (i.e., below 2 μm core diameter in Fig. 4a).

The nonlinear parameter $\gamma(\omega)$ of our LiCOF can be calculated from the intensity distribution of the fundamental mode, including the wavelength-dependent mode overlap and the wavelength-dependent nonlinear refractive indices of the core and cladding (see Methods) as depicted in Fig. 4d. For the ultrashort pulses used in our experiment (460 fs, 1.95 μm), the nonlinear refractive index is calculated to be $n_{2,CS_2} = 104 \times 10^{-16}\,\mathrm{cm^2W^{-1}}$ and the nonlinear parameter to be $\gamma = 0.28\,\mathrm{m^{-1}W^{-1}}$. The modal attenuation of our system is governed by the material absorption of CS$_2$ (Fig. 4e), which was estimated on basis of in-house measurements and previously reported data (see Methods). The absorption of the liquid is approximately four orders of magnitude larger than that of silica, at 2 μm wavelength. However, guidance along a few tens of centimetres of the CS$_2$/silica fibre is possible with transmission efficiencies well above 30% at this wavelength.

**Supercontinuum generation in liquid-core fibres**. To study the dynamics of HSWs, we generated supercontinua with a set-up that combines an ultrafast short-wave infrared laser source with a microfluidic system (see Supplementary Fig. 6). The pump source is a thulium-based fibre master oscillator connected to a fibre amplifier as described in[34], emitting pulses with a temporal width of $\tau_{FWHM} = 460$ fs at a repetition rate of 5.6 MHz and a wavelength centre at $\lambda_p = 1.95$ μm. It is coupled into a few-mode CS$_2$-LiCOF with a core diameter of 4.7 μm (Fig. 4c, $V = 4.55$ at $\lambda_p$), which allows operation at $\lambda_p$ within the anomalous dispersion regime close to the ZDW (1.83 μm, Fig. 4d).

In our experiment, we recorded the output spectra for increasing pump energies until the transmission drops because

of damage (see Supplementary Table 1). We observed substantial broadening of the output spectrum for increasing input pulse energy, with a maximum spectral extent from 1.1 to 2.7 μm at 14 nJ (Fig. 5a). Careful alignment ensured energy conversion within the fundamental mode across the entire bandwidth (see insets of Fig. 5a).

At a first glance, the spectral evolution follows the clean characteristics of classical soliton fission: after initial SPM, a sudden increase in the spectral bandwidth is observed at 2.5 nJ pulse energy, with a distinct short-wavelength shoulder (around 1.25 μm) neatly emerging far away from the pump $\lambda_p$. This point we denote as the supercontinuum onset. Increasing the pulse energy leads to an increased spectral bandwidth with more spectral fringes appearing, i.e., on the soliton side.

We numerically investigate the supercontinuum process with two types of nonlinear pulse propagation equation, namely a generalised and a hybrid form of the nonlinear Schrödinger equation. Owing to its novelty, the latter is discussed with the related results (Fig. 5c) at the end of this section. The simulations based on the general model (GNLSE, see Methods) shown in Fig. 5b clearly correlate with the experiments. In particular, the onset energy and the spectral location of the initial dispersive wave match very well, which confirms (i) an efficient coupling to the fundamental mode and (ii) an accurate balance between fibre dispersion and nonlinearity in the simulation. The further extent of the long-wavelength side of the spectrum, beyond the measured limit 2.7 μm, for higher pump energies might originate from missing parasitic effects not included in the simulation, such as self-focussing, thermal load, and optomechanical interactions.

The distinct intensity maxima on the short-wavelength side are associated with dispersive waves (i.e., Cherenkov radiation), which are phase-matched to the fundamental solitons shortly after the fission point[35]. With a phase-matching condition, we should be able to link the dispersive wave with the shortest wavelength (solid red line in Fig. 5a) to the most red-shifted maxima of the spectrum (dashed/dotted red lines in Fig. 5a), especially at pulse energies below 7 nJ where fission appears close to the fibre end and the first fission soliton is mainly unaffected by self-frequency shifts. The phase-matching condition includes a nonlinear phase term $\Delta\varphi_{NL}$, which is fundamentally different between instantaneous, non-instantaneous and hybrid systems (see Methods). Considering only an instantaneous nonlinear phase shift $\Delta\varphi_{NL}^i$ (case i in Fig. 5a) leads to an overestimation of the fission soliton wavelength and, thus, of the generated spectral bandwidth. The exclusive use of the non-instantaneous phase shift $\Delta\varphi_{NL}^{ii}$ (case ii in Fig. 5a) given by Conti et al. instead seems

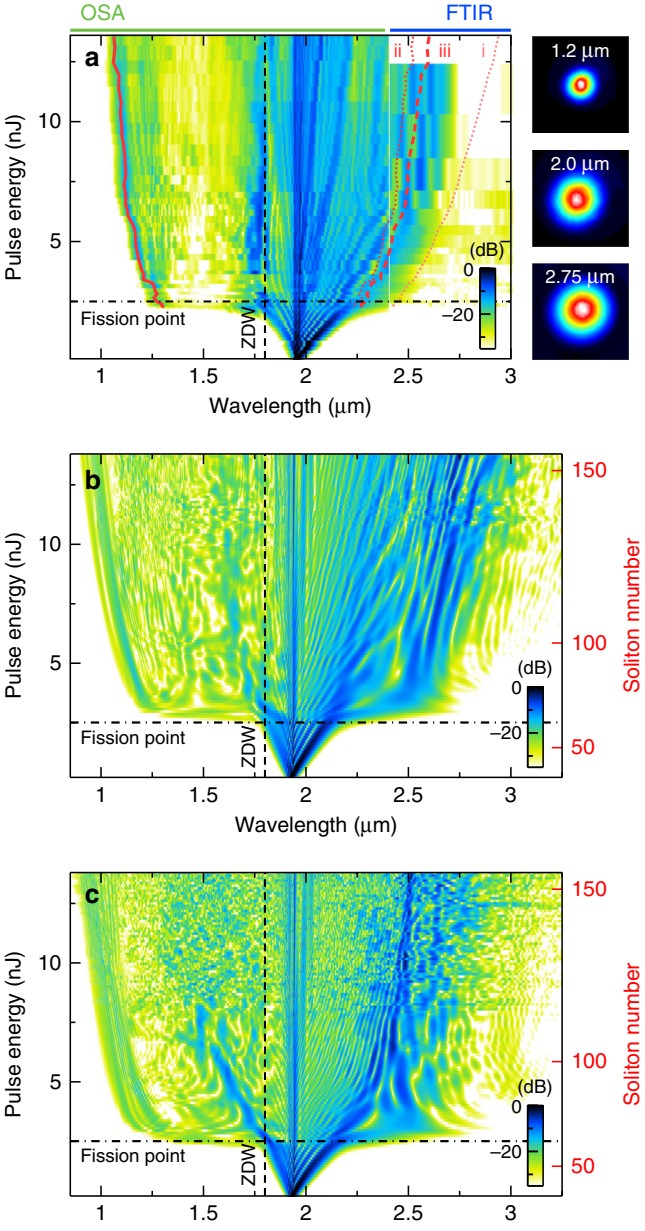

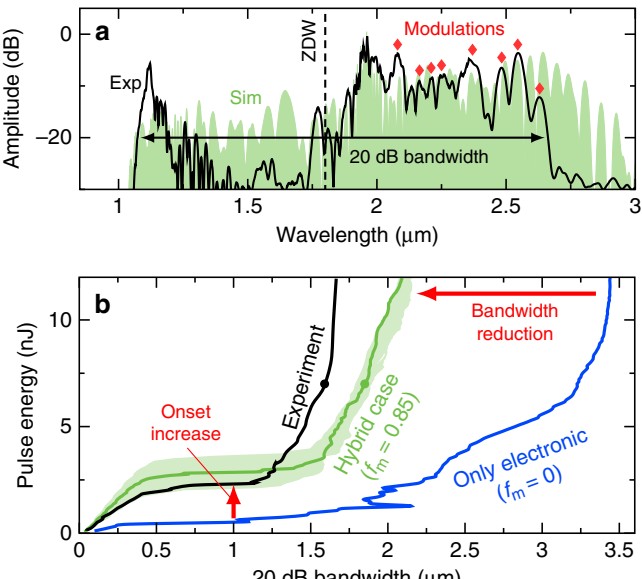

**Fig. 6** Indications for hybrid soliton dynamics. **a** Measured (exp) and simulated single-shot (sim) spectra at the output of a $CS_2$/silica LiCOF (fibre length: 7 cm, core diameter: 4.7 μm). The *red diamonds* mark a few locations of spectral fringes. ZDW zero-dispersion wavelength. **b** Measured and simulated bandwidth at 20 dB spectral contrast for increasing input energy in comparison to an equivalent fibre with entirely instantaneous response ($f_m = 0$). The *circles* label the positions of the spectra at 7 nJ shown in **a**. The *light green area* denotes simulations incorporating deviations of the non-instantaneous nonlinear response model and the pulse duration leading to $f_m = 0.85 \pm 0.04$ and $\gamma = 0.28 \pm 0.06$ m$^{-1}$W$^{-1}$

**Fig. 5** Comparison of experimental and simulated spectral fingerprints of generated supercontinua. **a** Measured output spectra (log. power scale: −35 to 0 dB) of the $CS_2$/silica LiCOF for increasing pulse energy. *Red lines*: measured spectral position of the dispersive wave with the lowest wavelength (solid) and the phase-matched wavelengths of the first fundamental solitary wave for the three different nonlinear phase shifts (numbers correspond to the cases discussed in the main text). The right-handed insets show the measured near-field profiles at three selected wavelengths (pulse energy 8 nJ). OSA optical spectral analyser, FTIR Fourier transform infrared spectrometer, ZDW zero-dispersion wavelength. **b** Simulated single-pulse spectra (GNLSE) for increasing input pulse energy (log. power scale: −35 to 0 dB), using a pulse shape reconstructed from the experiment, and the full temporal response of $CS_2$. **c** Simulated single-pulse spectra for increasing input pulse energy (log. power scale: −35 to 0 dB), using a specialised hybrid Schrödinger equation

to underestimate the spectral extent. A combination of both phase shifts weighted by the molecular fraction $f_m$ (case iii in Fig. 5a), which we denote as hybrid nonlinear phase

$\Delta\varphi_{NL}^{iii} = (1 - f_m)\Delta\varphi_{NL}^{i} + f_m\varphi_{NL}^{ii}$, matches the experiment best. Hence, our system does not evolve classically.

Remarkably, we observed clean soliton fission as the dominant broadening process at soliton numbers much larger than the coherence limit (i.e., $N \gg 10$). The observation of fine spectral fringes between 2.2 and 2.7 μm (e.g., red diamonds in Fig. 6a) at such large soliton numbers and the absence of signatures of modulation instabilities at the fission point indicate a high pulse-to-pulse stability, which is in accordance to our previous analysis in Fig. 3. The modulation contrast of the fringes on the soliton side are also in the order of 5–10 dB and matches sufficiently well to noise-free simulations, e.g., in Fig. 6a. All lines of evidence support our theoretical understanding of the noise dynamics and allow the prediction of a high first-order coherence in this system (see Fig. 3).

We find further indicators of new soliton dynamics in the bandwidth of generated spectra and the supercontinuum onset energy. In Fig. 6b we compare the spectral 20 dB bandwidth of the measured spectra with simulations of an entirely instantaneous system (case i: $f_m = 0$) and the hybrid $CS_2$ system (case iii: $f_m = 0.85$). The pure electronic system (case i) shows a large bandwidth increase, up to 3.5 μm, which is only limited by distinct material absorption lines. The hybrid system (case iii) shows higher supercontinuum onset energy and significantly reduced bandwidth, both in good agreement with the experimental data. The error margin of the calculated bandwidth was estimated by assuming the largest deviations of the nonlinear response given by Reichert et al.[16] and measurement inaccuracies of the pulse width of about 9% (±40 fs). Both observables indicate the dynamics of HSWs in accordance with our simulations in Fig. 2.

The experimental evidence for the applicability of the theory by Conti et al. motivates us to look for a specialised form of the GNLSE for our hybrid system based on their model. By a special treatment of their non-instantaneous phase term (see Methods), we are indeed able to obtain a hybrid Schrödinger equation of the form

$$\partial_z\widetilde{A}(z;\omega)+\frac{1}{2}\alpha_{\mathrm{m}}(\omega)\widetilde{A}-i\overline{\beta}(\omega)\widetilde{A}=$$
$$i\widetilde{\gamma}(\omega)\ \mathcal{F}\big\{A(z;t)\big[(1-f_{\mathrm{m}})|A(t)|^2+f_{\mathrm{m}}V_0(t)\big]\big\}, \tag{2}$$

With the spectral pulse envelope $\widetilde{A}(z;\omega)$ and its time-domain counterpart $A(z;t)$, the modal attenuation $\alpha_{\mathrm{m}}$, the modal dispersion $\overline{\beta}$, and the modified nonlinear parameter $\widetilde{\gamma}$ (see Methods). All fibre parameters are included with their full dependence on (angular) frequency $\omega=2\pi c_0\lambda^{-1}$, which may be reduced for systematic studies in future work. The non-instantaneous phase acts as an additional linear potential $V_0(t)$ weighted by the molecular fraction $f_{\mathrm{m}}$.

Figure 5c shows the simulation results using our new model equation for increasing input power. The model describes the spectral characteristics of the experimental data remarkably well, especially around the supercontinuum onset. The fission point appears at the same energy level, and spectral features like dispersive waves and the fine spectral fringes on the long-wavelength side are well captured. The spectral bandwidth of the simulation is slightly overestimated compared to the experiment, but matches well with the more general GNLSE. The new model starts to become inaccurate only for long fibre lengths (i.e., $L\geq L_D$) and large soliton numbers (here $N>100$), since strongly delayed wave packets temporally shift out the initial potential which is not physical.

However, because of the good match between the hybrid model and both general model and experiment, the specialised Schrödinger equation promises to be a quick tool to evaluate hybrid systems. It also forms a strong link to the previous theory and might evolve to become a key tool to further understand hybrid soliton dynamics in future studies, especially around the fission point.

## Discussion

In this contribution, we report on the theoretical and experimental evidence of unexplored HSWs as a result of highly non-instantaneous nonlinear interactions in LiCOFs. The solitary states show unique signatures, such as asymmetric pulse shapes and strong robustness against phase noise, and have profound impact on the characteristics of soliton-based supercontinua, with the outstanding feature of perfect shot-to-shot coherence across the entire generated bandwidth. The investigated fibre system was composed of a CS$_2$/core with a diameter of 4.7 μm and a silica cladding and pumped in the anomalous dispersion regime by a 460 fs fibre laser operating at 1.95 μm. Octave-spanning supercontinua between 1.1 and 2.7 μm were measured at 14 nJ pump energy. Compared to media with purely instantaneous Kerr response, the supercontinua show very fine spectral fringes with overall reduced bandwidths, which are both associated with the emergence of HSWs. Nonlinear pulse propagation simulations using both the generalised and a hybrid nonlinear Schrödinger equation match well in bandwidth and spectral features to the measurements, and explain the characteristics of the supercontinua with the appearance of the hybrid states in a temporally linear potential created by the non-instantaneous response.

Although the presented CS$_2$/silica system is only one example, we are convinced that the hybrid nonlinear Schrödinger equation is also applicable to other highly non-instantaneous waveguides, as it forms the first strong link between the non-instantaneous

solitons theoretically predicted by Conti et al. and the states, which are observable in realistic hybrid–nonlinear systems. The fact that the hybrid nonlinear Schrödinger equation describes the observed soliton dynamics very well suggests that the non-instantaneous nonlinear phase plays a major role already in the soliton fission process. In future studies, this Schrödinger equation might help to answer fundamental questions, such as whether the new type of solitary wave actually appears immediately during fission or whether the hybrid nature of those states is imposed on classical solitons during propagation after the actual fission process.

Supercontinuum generation in liquid-core fibres is currently the only tool to study non-instantaneous solitons, since a direct investigation of low-order solitons is prevented by optical attenuation. Only accurate dispersion tuning within a low-loss window of the investigated liquid with selectively filled microstructured fibres might allow further insights, which, however, is still a challenge in terms of both knowledge about the optical properties of liquids and sample fabrication. Besides fundamental studies in nonlinear physics, soliton excitation in liquid-core fibres can pave the way for novel spectroscopic applications and coherent and tuneable mid-infrared light sources. Liquid-core fibre supercontinuum sources potentially offer an attractive alternative to other highly coherent approaches like all-normal dispersion broadening[36] or externally seeded soliton-based supercontinuum generation[37]. Future studies will reveal the potential of hybrid-soliton-based supercontinuum generation as highly coherent broadband light sources, particular from the perspective of pulse-wise spectral stability and wavelength tunability.

## Methods

**Liquid-core fibre dispersion design**. The dispersion design of our LiCOF requires precise knowledge about the refractive index dispersion of CS$_2$. The existing dispersion models[38, 39], however, do not account for the strong absorption of CS$_2$ at 6.6 μm. As a consequence, these models provide an incomplete description of the spectral distribution of the GVD, particularly at the mid-infrared wavelengths. Here, we use additional refractive index data published in previous work to obtain a new two-term Sellmeier model (at 20 °C, see Supplementary Fig. 1 for details):

$$n_{\mathrm{CS}_2}^2-1=\frac{1.499426\lambda^2}{\lambda^2-(0.178763\,\mu\mathrm{m})^2}+\frac{0.089531\lambda^2}{\lambda^2-(6.591946\,\mu\mathrm{m})^2}, \tag{3}$$

with the vacuum wavelength $\lambda$ in μm. The denominator of the second term includes the long-wavelength resonance, whereas the constants of the first term differ only slightly from those reported by Kedenburg et al.[39].

With Eq. 3 and the refractive index dispersion of silica[40], we calculate the GVD of the LiCOF by numerically solving the transcendental dispersion equation of a cylindrical step-index fibre[9].

The nonlinear parameter $\gamma$ is calculated using[41]

$$\gamma(\omega)=\frac{\omega}{c_0}\frac{n_{2,\mathrm{CS}_2}\iint_{co}S_z^2 r\mathrm{d}r\mathrm{d}\varphi+n_{2,\mathrm{SiO}_2}\iint_{cl}S_z^2 r\mathrm{d}r\mathrm{d}\varphi}{\big(\iint_\infty S_z^2 r\mathrm{d}r\mathrm{d}\varphi\big)^2}\approx\frac{\omega n_{2,\mathrm{CS}_2}}{c_0 A_{\mathrm{eff}}}\equiv\widetilde{\gamma}(\omega)\cdot\frac{1}{A_{\mathrm{eff}}^{3/4}}, \tag{4}$$

where $n_{2,\mathrm{CS}_2}$ and $n_{2,\mathrm{SiO}_2}$ are the total nonlinear refractive indexes of CS$_2$ and silica, respectively, and $S_z$ the z-component of the Poynting vector of the fibre mode. The core nonlinearity is by two orders of magnitude larger than that of the cladding ($n_{2,\mathrm{CS}_2}\gg n_{2,\mathrm{SiO}_2}$), whereby the latter can be neglected for simplicity. To account for the dispersion of the mode field area[42], we isolate the factor $A_{\mathrm{eff}}^{-3/4}$ from the conventional definition of $\gamma$, which serves as normalization factor of the field amplitudes in the nonlinear term of the Schrödinger equation (e.g., see Eq. (6)).

The total nonlinear refractive index of CS$_2$ is calculated to be[16]

$$n_{2,\mathrm{CS}_2}=n_{2,\mathrm{el}}+\frac{\int I(t)\int R(t-t')I(t')\mathrm{d}t'\mathrm{d}t}{\int I(t)^2\mathrm{d}t}=n_{2,\mathrm{el}}+n_{2,\mathrm{m}}, \tag{5}$$

with $n_{2,\mathrm{el}}$ as the nonlinear refractive index originating from the instantaneous electronic contribution, which is calculated using $n_{2,\mathrm{el}}=3\chi^{(3)}(3\varepsilon(\omega)\varepsilon_0 c_0)^{-1}$ with the electric and the relative permittivity $\varepsilon_0$ and $\varepsilon$, and the speed of light $c_0$. The convolution between the nonlinear optical response function $R(t)$ and the temporal intensity distribution $I(t)$ of the incoming optical pulse makes the nonlinear refractive index dependent on the pulse duration. A detailed description of the $R(t)$ model of CS$_2$ used for our calculations can be found in the work by Reichert et al.[16].

| Fig. | $\alpha_0$ (m$^{-1}$) | $\beta_2$ (10$^4$ fs$^2$) | $\beta_3$ (10$^5$ fs$^3$) | $\beta_4$ (10$^5$ fs$^4$) | $\beta_5$ (10$^5$ fs$^5$) | $\gamma_0$ ((W m)$^{-1}$) | $\gamma_1$ (fs (W m)$^{-1}$) |
|---|---|---|---|---|---|---|---|
| 1a | 0 | −2.0505 | 0 | 0 | 0 | 0 | 0 |
| 1b | 0 | −2.0505 | 0 | 0 | 0 | 0.0412 | 0 |
| 1c,d | 0 | −2.0505 | 0 | 0 | 0 | 0.2823 | 0 |
| 2 | 0 | −2.0505 | 3.8389 | −4.3966 | 2.3937 | 0.2823 | 0.4059 |
| 3 | 2.7187 | −2.0505 | 3.8389 | −4.3966 | 2.3937 | 0.2823 | 0.4059 |
| 5 | 2.7055 | −2.0125 | 3.8263 | −4.3767 | 2.3829 | 0.2698 | 0.3874 |

**Table 1 Simulation parameters.** Best approximation of the broadband fibre parameters (at $\lambda_p = 1.95\ \mu m$) used in the simulations of this paper

**Liquid-core fibre filling and light coupling**. A liquid-core step-index fibre can straightforwardly be fabricated by filling a capillary with the solvent using capillary force[43], substantially reducing the fabrication effort compared to selectively filled photonic crystal fibres. Our capillaries were fabricated in-house. Two home-built opto-fluidic fibre mounts were used to supply the solvent via two fluidic side-ports to the capillary ends, each centred in the middle fibre port facing a Sapphire window. These mounts allow simultaneous filling and exchange of the $CS_2$ while launching the ultrafast pulses into the liquid-core. The flow was controlled with a liquid chromatography pump. Initially, the mounts were subsequently flushed with a delay of a few minutes to wait for the complete filling of the capillary by capillary force.

Light was coupled in and out of the fibre using aspheric lenses with high numerical aperture (NA) (input lens NA: 0.3, output lens NA: 0.68). The coupling was optimised at power levels of a few milliwatts, where no significant spectral broadening is observed. Since the LiCOF is slightly multimodal, efficient excitation of the fundamental mode was ensured by imaging the output mode patterns with a thermal camera (MCT detector, FLIR SC7000) while coupling optimization is done. Coupling efficiencies on the order of 50% are reached, taking into account a modal attenuation of 14.5% along the 7 cm-long sample and reflections at various interfaces. The coupling is stable over several days under atmospheric pressure and even while flushing the optofluidic mounts with flow rates up to 10 ml min$^{-1}$. The set-up was kept under pressure, between 4 and 6 bar (at 20 °C), to avoid bubble formation as reported earlier[32, 33], which in fact turned out not to be crucial for the pump wavelength considered here.

**Supercontinuum measurements and data processing**. The output light was measured behind the sample with an InF$_3$ multimode fibre centred in the collimated output beam and connected either to optical spectral analysers (optical spectral analyser (OSA), Yokogawa, spectral range: 0.7–2.4 μm) or to an externally coupled Fourier transform infrared spectrometer (Jasco FTIR-6300, Type A, TGS detector). By using average integration times between 8 and 10 min, the Fourier transform infrared spectrometer (FTIR) reached a dynamic range of 30 dB. The power at the input and output of the fibre was measured with a thermal power meter before and after each spectral recording (see Supplementary Fig. 7), confirming that nonlinear absorption plays a minor role in our investigation.

The recorded spectra were weighted with a calibration curve gained from a power measurement with a series of band-pass filters (1.2–2.75 μm). Two spectra gained from the OSA and the FTIR at identical input power levels were combined by subtracting an offset from the overlapping FTIR spectrum at the long-wavelength edge of the OSA (2.4 μm). The spectra were normalised individually to the in-fibre pulse energy.

**Generalised Schrödinger equation**. For our simulations, we used a GNLSE of the following form:

$$\partial_z \widetilde{A}(z;\omega) + \frac{1}{2}\alpha_m(\omega)\widetilde{A} - i\bar{\beta}(\omega)\widetilde{A} = i\tilde{\gamma}(\omega)\ \mathcal{F}\left\{ A(z,t)\int_{-\infty}^{\infty} h(t')|A(t-t')|^2 dt'\right\},$$ (6)

with the spectral field envelope $\widetilde{A}(z;\omega)$ and its time-domain counterpart $A(z;t) = \mathcal{F}\{\widetilde{A}(z;\omega)A_{\text{eff}}^{-1/4}(\omega)\}$ normalised to the effective mode area $A_{\text{eff}}$[42], the general response function $h(t)$ (normalised using $\int_{-\infty}^{\infty} h(t)dt = 1$), the propagation constant in the moving time frame $\bar{\beta}(\omega) = \beta(\omega) - \beta(\omega_0) - \omega\partial_\omega\beta|_{\omega_0}$ (with $\omega_0 = 2\pi c_0/\lambda_p$), the modal attenuation coefficient $\alpha_m$ and the modified nonlinear coefficient $\tilde{\gamma}$ from Eq. (4). The operator $\mathcal{F}$ denotes the Fourier transformation. No noise is added if not stated otherwise. The GNLSE was solved with a split-step Fourier transform method where the nonlinear step was solved with a Runge-Kutta integrator of the 4th order.

The general response function $h(t)$ includes both the electronic nonlinear contribution (assumed to be instantaneous using Dirac delta function) and the normalised nonlinear molecular response function, leading to $h(t) =$ $(1-f_m)\delta(t) + f_m R(t)/\int R(t)dt$. The weight between the two contributions is the molecular fraction factor $f_m$.

For simulations of the experiment (Fig. 5), the complex field envelope was reconstructed from the measured spectrum of the pump laser using a third-order dispersion phase offset of $\beta_3 = -0.025$ ps$^3$ estimated from the overlap with the measured autocorrelation.

The modal attenuation of the fundamental mode $\alpha_m$ (Fig. 4c) was calculated using the bulk material absorptions weighted by the overlap integrals of the propagating mode with the core and cladding, respectively. The spectral distribution of the $CS_2$-bulk absorption in the mid-infrared is estimated by a multi-Lorentz fit on the data of previously reported absorption measurements[44, 45]. We measured the absorption of $CS_2$ at near-infrared and visible wavelengths using a metre-long metal tube (diameter 12 mm) with plain sapphire windows positioned in the collimated probe beam of a broadband light source (NKT Photonics SuperK). The transmitted light was guided to an optical spectrum analyser via a metre-long multimode silica fibre. The recorded absorption values around 1.95 μm match well to data reported in earlier works[45]. The absorption of $CS_2$ at shorter wavelengths is very low and thus could not be retrieved in our experiments due to the limited dynamic range of the used spectrometer (see Fig. 4e). Therefore, we approximated the absorption below 1.85 μm by a linear dependence of $\alpha_m$ on $\lambda$. This extrapolation is important for the consistency of the numerical simulations (only continuous functions are used), but has no influence on the simulation results as the overall modal attenuation for wavelength shorter than 1.8 μm is very low. Since the absorption of bulk $CS_2$ is orders of magnitude higher than that of silica[46], the modal loss is governed by the absorption of $CS_2$ across the entire bandwidth of interest. For instance, the fundamental HE$_{11}$ mode has an attenuation of 0.12 dB cm$^{-1}$ at 2 μm wavelength.

In Table 1, we provide the $\beta_n$ and $\gamma_n$ parameters, which give the best fit to our calculated dispersion and nonlinear parameter to make the simulations more easily accessible. Note that the results might deviate from our broadband model. The assumed pulse shape in the simulations for Fig. 1 to Fig. 3 is a sech-squared shape with a full width at half maximum of $\tau_{\text{FWHM}} = 450$ fs ($\tau_0 = 255$ fs).

**Specialised Schrödinger equation**. The GNLSE in Eq. 6 involves the general response function $h(t) = (1-f_m)\delta(t) + f_m R(t)/\int R(t)dt$ within a convolution integral, which can only be solved analytically under certain assumptions. The first term results in the well-known instantaneous Kerr term $\gamma_0|A|^2$. The highly non-instantaneous molecular contribution can be approximated by a static linear potential $\gamma_0 \mathcal{E}_p R(t)$[17]. However, this potential causes numerical problems arising from the discontinuity at $t = 0$ implied by causality in $R(t)$. As a solution, we replace $\mathcal{E}_p R(t)$ by a static initial potential $V_0(t) = \int |A(0;t')|^2 R(t-t')dt'$, which is the convolution of the nonlinear response function with the input pulse intensity at $z = 0$. This yields the hybrid Schrödinger equation as given in Eq. 2, in which electronic and molecular contributions are again weighted by the molecular fraction $f_m$. Unlike the GNLSE, the convolution between the response and pulse intensity has to be calculated only once for the initial pulse, which yields a quasi-static contribution to the potential and is numerically less demanding.

**Nonlinear phase-matching conditions**. The phase-matching condition between a dispersive wave and a fundamental soliton after fission is $\beta(\omega_{\text{DW}}) - \omega_{\text{DW}}\beta_1(\omega) = \beta(\omega) - \omega\beta_1(\omega) + \Delta\varphi_{\text{NL}}$, with $\beta$ being the propagation constant, $\omega_{\text{DW}}$ the angular frequency of the dispersive wave, and $\beta_1 = \partial_\omega\beta$ the group velocity. The classical nonlinear phase shift is $\Delta\varphi_{\text{NL}}^i = \gamma_0 P_0(2N - 2j + 1)^2/N^2$ with the maximum peak power of the $j$-th soliton[47]. In our calculations for Fig. 5a, we have $j = 1$ since we only consider the first fundamental soliton after fission, which creates the most blue-shifted dispersive wave. The non-instantaneous nonlinear phase shift defined by Conti et al. is $\Delta\varphi_{\text{NL}}^{ii} = \gamma_0 \mathcal{E}_p \tilde{R}(\omega)$, with the pulse energy $\mathcal{E}_p$ and the Fourier transform of the molecular response function $\tilde{R}(\omega)$.

**Impact of model uncertainties**. The spectral bandwidth of the generated supercontinua crucially depends on the GVD and molecular fraction factor, which inherently underlie experimental uncertainties such as insufficient knowledge of material dispersions or of the spectral distribution of the nonlinear response of the

liquid. Here, we explain the impact of the individual uncertainties on our simulation results.

*Material dispersion.* To analyse the impact of the GVD (in particular, the refractive index dispersion of the core material) on the supercontinuum bandwidth, we simulate supercontinuum spectra for three different LiCOFs, all having silica claddings and equal core diameters but different models for the $CS_2$ refractive index dispersion under identical pulse input conditions ($\tau_{FWHM} = 460$ fs, $\lambda_p = 1.95$ µm). Besides the single-term Sellmeier dispersions reported by Samoc et al.[38] and Kedenburg et al.[39], we included our double oscillator model (Equation 3), showing a distinct difference to the single oscillator models for $\lambda > 2$ µm (Supplementary Fig. 1). Using our two-term Sellmeier equation results in the best-match scenario to the experiment in terms of bandwidth and supercontinuum onset energy (Supplementary Fig. 2a), revealing Eq. 3 to be an accurate description of the dispersion of $CS_2$ in the mid-infrared. Furthermore, we repeated the simulation for our model with three different core diameters (4.4, 4.7 and 5.0 µm) to demonstrate the weak influence of dispersion uncertainties on the spectral bandwidth (Supplementary Fig. 2b). The supercontinuum onset, however, shows a stronger dependency on the dispersion.

*Nonlinear refractive index.* The most recent model of the nonlinear refractive index of $CS_2$ has been included in our work[16], whereas the reported measurements come with rather large error margins. Even small deviations on the response function influence the nonlinear refractive index and, thus, the molecular fraction $f_m$ and the supercontinuum spectra. For example, a small correction of the response model of $CS_2$ published in the erratum by Reichert et al.[48] decreased the supercontinuum bandwidth and gave a better match to experiments, as it increased the molecular fraction from $f_m = 0.81$ (old model) to $f_m = 0.85$. Additional simulations with artificially varied $f_m$ reveal the strong influence of this parameter on bandwidth, fission length and noise characteristic, especially for $f_m > 0.7$ (Supplementary Figs 3 and 4). In this context, the good match between our simulations and experiment in Fig. 5 without having applied any fit parameters highlights the model quality of the fibre properties used in the simulations, as well as it confirms the validity of our approach to reveal new soliton dynamics.

*Unknown losses.* Further limitations on the bandwidth of the generated supercontinua are the additional linear losses at mid-infrared wavelengths or nonlinear losses. A direct comparison of the input-to-output power characteristics of our system (see Supplementary Fig. 7) shows a good match between measurements and simulations, and no evidence for an unknown dominant source of loss. This also justifies neglecting nonlinear losses in our model.

**Data availability**. The data that support the findings of this study are available from the corresponding author upon request. The raw data of each figure in this manuscript are accessible via the following link: http://dx.doi.org/10.6084/m9.figshare.4816462.

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

## Acknowledgements

We thank Dr Fabio Biancalana and Dr John C. Travers for fruitful discussions and hints on critical points of this study, and Dr Alessandro Tuniz for a critical review. We thank Dr Wonkeun Chang and Professor John M. Dudley for important feedback on our coherence calculations. We thank Franka Jahn from the IPHT Jena for providing electron micrographs of our capillaries used in this work. M.C. and M.A.S. acknowledge financial support from the Deutsche Forschungsgemeinschaft (DFG) via the grants SCHM 2655/2-1 and SCHM 2655/3-1, and the Thuringian State (projects 13023-715, 2015FGI0011, 2015-0021) partly supported by the European Regional Development Funds (ERDF). C.G. and F.S. acknowledge support by the Carl-Zeiss-Stiftung. M.G. acknowledges financial support by the Research School of Advanced Photon Science of the Helmholtz Institute Jena.

## Author contributions

M.C.: Designed and fabricated the fibre sample, conducted the experiments and performed the simulations. J.K.: Fabricated the capillaries. M.G., C.G. and F.S.: Designed the fibre laser system under the supervision of J.L. and A.T. The original idea was conceived by M.A.S., who also supervised the research. M.C. and M.A.S.: Prepared the manuscript. All authors contributed to discussions and the interpretation of the results.

## Additional information

**Competing interests:** The authors declare no competing financial interests.

