## [Peer Review file · Nature Communications]

Editorial Note: the text of this manuscript has been modified to ensure that all work is attributed correctly.

Reviewers' comments:

Reviewer #1 (Remarks to the Author):

The authors present an experimental study of supercontinuum generation in a fiber with a CS₂ liquid core. The size of the core has been carefully chosen to access the anomalous dispersion regime so that the continuum is governed by soliton dynamics. The supercontinuum that is generated covers a wavelength range of $\sim 1-2.7\mu\text{m}$, which does appear to be the largest continuum reported to date in this liquid (though not by a significant amount as Ref. [23] reports a continuum from 1.2-2.4 μm). However, the main claim of this work is not the continuum itself, but the observation of a new kind of non-instantaneous soliton state that arises due to the delayed Raman response of the liquid.

The discovery of new pulse solutions is always of fundamental interest, and the authors make a case that such nonlocal soliton dynamics will allow for the generation of more coherent continuum sources, which would certainly be useful. However, my main concern with this paper is not related to the novelty of these solutions, but to how convincing their evidence is for their observation. From what I understand there are two distinguishing features of their spectrum that point to the excitation of the nonlocal solitons. The first is the reduced bandwidth of the continuum. On this point, how can they be sure that this isn't due to any additional losses in their system, or indeed any deviations in the dispersion properties of the fibers from their predictions (i.e., it is easy to imagine that movement and/or thermal gradients of the liquid core could result in variations in the refractive index)? The second appears to be based on the fine spectral features seen in Fig. 3(a) (identified by the diamonds). However, this spectrum is not compared with the simulations or anything else that could confirm the appearance of the solitons, and so there seems to be a lack of supporting evidence for this claim. Thus before this paper is accepted it would be helpful to see some stronger evidence presented for the observation of these non-instantaneous soliton solutions. I also have a few other comments and suggestions for the authors, as outlined below.

1. The authors refer to these non-instantaneous soliton solutions as "linearons". From what I can tell this is the first published use of this term and thus perhaps some explanation is required. I would say that I am not a great fan of this term myself, but it is up to the authors to decide if they want to introduce it to the community.

2. In my opinion Fig. 1(c) is too small for the amount of information it contains. Also, the authors talk about working close to the first zero dispersion wavelength (ZDW), however, I can only see one ZDW in the dispersion plot of Fig. 1(c) - not two. They need to explain this better, and expand the plot if necessary.

3. I have no idea what message they are trying to convey with the statement below: "From the experimental point of view, domain (i) is critical in terms of coupling, guidance and fiber nonlinearity due to a weak field confinement within the nonlinear core."

4. The authors use wavelength dependent values for their material parameters, however, the values for n_2 in Ref. [13] only cover from 500-1500nm. How do they estimate the values beyond this? There seems to be much emphasis on the model for the linear refractive index, and not so much on n_2 .

5. It is not clear to me what causes the drop in power due to high peak powers. For high average powers I understand that there is damage to the material, but I can't see an explanation for the case of high peak powers. Is the drop in power reversible or is this also due to some permanent change in the system?

6. Fig. 3 compares the experimentally generated supercontinuum with their numerical model, generally showing a reasonable agreement. However, it would be helpful to show a direct

comparison between the experiments and the model for one input energy to back up their statements about the appearance of the "fine features" that define the soliton behavior (as discussed on page 8). This could simply be achieved by overlapping the simulated spectrum in the top part of (a). On this note, can the authors comment on why such distinguishing spectral features have not been seen in the previous continuums generated in this material, which have also been pumped with femtosecond pulses? Related to this point, can the authors also show an evolution plot for the propagation in the temporal domain obtained from the model? Their previous paper (Ref. [9]), where these solitary pulses were proposed, largely focuses on the temporal features of these pulses and thus it would be good to know if any distinguishing features can be seen in this domain.

7. In Fig. 3(a), I am not sure that the mode profiles are enough to claim single mode excitation. Have they consider S^2 measurements and/or at least fitting the expected profiles with the fundamental mode? How many modes would they expect this fiber to support given the core size and the index of CS₂? Some further comment to back up this statement would help.

8. On page 8 they have a discussion about the non-solitonic radiation (blue side) being bound to the solitons on the red side of the spectrum. Again, it would be helpful to see some temporal plots which show evidence of soliton formation to directly map their appearance to the spectral features. This is fairly common in papers reporting soliton-induced supercontinuum.

9. On pages 11-12 the authors discuss the coherence properties of the generated continuum where they make the claim that the linearon-induced continuum is more coherent than a continuum generated from an instantaneous nonlinearity. This section would be strengthened if the authors could conduct the experiments as well. For example, coherence measurements have been performed over a wavelength range of 1500-1800nm in the paper: Leo et al. Opt. Exp. v.22., p.28997 (2014) and something similar could be done here.

10. At the top of page 13 the authors discuss some numerical simulations conducted in fibers using different Sellmeier models, however, I cannot find where these results are reported. Given that these are used to support their claim that the two-term Sellmeier is critical it would be helpful to see these somewhere, even if in the supplementary information. Similarly, the results of the modelling with different molecular fractions in gamma could also be shown in the supplementary information.

11. In the conclusion they claim that one of the advantages of their system is that they don't need to use complex PCFs to generate a flat SC, but then say that they could over problems with the absorption of the liquids by using PCFs. These statements seem to cancel each other out.

12. There are a number of typos and grammatical errors in the manuscript and references. I would recommend that the authors proofread this carefully before resubmitting.

13. Some of the references seem a little redundant. There are 5 for the refractive index of CS₂, and several of the reports seem to overlap in values and wavelength.

Reviewer #2 (Remarks to the Author):

Review of “Non-instantaneous soliton dynamics within mid-infrared supercontinuum generated in liquid-core fibers”

SUMMARY

The authors present a study of non-instantaneous nonlinear optical dynamics inside a fiber waveguide filled with liquid. The target is to show so-called ‘linearons’ as established in Conti PRL 2010. The authors clearly state that this work represents an ‘indication’ of linearons in lieu of direct observation, which is a reasonable approach given what is known about the relationship between classical solitons and supercontinuum. The topic certainly seems worth pursuing, especially given the ability to generate coherent broadband beams. It should be emphasized that these are sound experiments. The challenge of even having access to the light sources and technology required to do the experiment is commendable.

Unfortunately the experimental evidence as well as the accompanying analysis falls far below the standard expected for a high impact paper and cannot be recommended for publication in Nature Communications.

The technical description has a number of deep flaws and does not currently meet the standard for publication in lesser journals. In particular, the manuscript fails to properly analyze the experiments following the criteria and formalism outlined in Conti PRL. This is especially true of the analysis of a soliton number and non-soliton radiation, which are misleading as will be outlined below. The conventional soliton world does not apply here. That said, this topic is new and it is apparent that this analysis is challenging. The authors will be able to correct this part with some effort.

One of the major claims of novelty is the use of anomalous dispersion (AD). Importantly, reference 42 appears to lay the foundation for this work and is left to only a brief mention at the very end of this paper. Advance over prior work would need to show a significant advance in understanding.

The broadening analysis presented is a reasonable first approach. However, it could have a number of origins besides linearons, namely nonlinear absorption, uncertainty in the experimental dispersion, or even inhomogeneity of dispersion and nonlinearity along the waveguide. These different mechanisms should be considered. Coherence is another potential piece of evidence. However, no measurements demonstrating coherence are demonstrated here.

The paper will not significantly influence the field. A direct proof of linearons (or linearon supercontinuum) for example, a coherence measurement (in lieu of simulation), along with convincing analysis that the NSR could only come from linearon dynamics is required for a convincing demonstration. The analysis should also rule out other effects.

TECHNICAL POINTS (MAJOR)

(1) Definitions for the linearon and use of conventional soliton definitions

- The current manuscript incorrectly applies the traditional definition of soliton number to the linearon situation.

The traditional definition of the soliton number $N^2 = \gamma * P * T_0^2 / \beta_2$ comes from the Kerr nonlinearity with GVD in a dimensionless nonlinear Schrodinger equation. This is not the case with linearons and applying that definition to the situation here does not make sense. (p8).

The definition of the linearon number is given as N_{script} in Conti PRL in which they explicitly state: “*The constant N cannot be written explicitly*, and is found by requiring that the total soliton energy is E.”

I understand that the manuscript attempts to lean on collective knowledge as a means to understand the system. However the application of these definitions is misleading and incorrect.

- Linearon shape is not a hyperbolic secant.
Thus the traditional soliton definitions do not apply. Related to this, the definition of full-width at half-maximum is given for a pulse of hyperbolic secant form. However, this is not the shape of a linearon as described in Conti PRL.
- Side note: For a conventional soliton system N refers to the *injected* soliton number. This is in contrast to the comment on p.8: “*refers to the number of solitons appearing after the fission point.*”

All comments mentioning the soliton number should be re-visited.

(2) Pure linearon and linear losses

One wonders why pure linearons are not possible here. The manuscript states: “*However, the absorption of CS₂ does not allow low-loss guidance along several meters of fiber thus preventing a direct proof of linearons.*”

However, Reference 23 has experimental measurements showing the CS₂ is low loss except at 2.2 μm . Did the authors measure this value directly for their own? How do we reconcile this difference?

Perhaps there is also a technical limitation to the liquid fiber length, in terms of homogeneity or related.

(3) Non-solitonic radiation

- The manuscript appears to use the classic NSR definition. However, the conditions for non-solitonic radiation are different for linearons than conventional solitons as outlined in Conti PRL 2010:

“The energy- dependent part is an extra contribution to the resonant condition that is unique for highly noninstantaneous solitons, and allows us to tune the frequency position of the emitted radiation by adjusting the total input pulse energy.”

This should be quantitatively described in this paper as part of the proof of linearons. Given that it is different NSR, any time that it is mentioned in the text it should be highlighted as this is quite different to conventional knowledge.

- *P6 – “The spectral evolution is characteristic for soliton-driven SCG: After initial self-phase modulation (SPM) a sudden increase of the spectral bandwidth at 2.5nJ is observed with a distinct short- wavelength shoulder (around 1.25 μ m) appearing far apart from λ_p .”*

COMMENT: This suddenly emergent blue shoulder is typically associated with NSR and arises because the soliton bandwidth (at this narrow temporal width) overlaps with the linear phase matching of the NSR and seeds it with energy. The description here leads us to think it is fission itself and should be clarified.

- On the positive side, it should be possible for the authors to use their spectral results in Fig. 3 to compare this with calculations, and see if their observation of NSR matches that predicted for linearon dynamics. This would fit well at the top of page 8 where they say ‘we can show’. This could go in the supplement (or even a specialized paper) where you describe the specific forms of these terms.
- *“The prominent red shoulder of the SPM spectra, which is also visible in the measurements, denotes strong temporal self-compression (self-steepening [25]) of the optical pulse.”*
 - Comment: Red shoulders are typically due to the Raman effect. Another possibility is asymmetric SPM from some perturbation or loss. Ref. 42. Claims theirs in CCl₄ comes from Raman, though one would need to re-visit their simulations to verify this is indeed true. Conti discusses the suppression of Raman under certain conditions. Does your system meet those conditions? If not, what is the explanation for the red-shoulder? Did you attempt a minimized simulation with just the linearon and self-steepening to see if this red shoulder could come from that?
 - Strong temporal self-compression arises from higher-order soliton compression, as opposed to the self-steepening mentioned here. Separately, in Conti PRL, they explicitly state (p1,c2) that the shock term (i.e. self-steepening) is neglected. That said, the authors could prove them to be incorrect by showing simulations with and without this term, or showing an analytic reason.
- *“Here, the higher-order soliton, which in fact is the initial input pulse, breaks up into multiple fundamental solitons on the long wavelength side and sheds energy towards shorter wavelength via NSR [5, 25].”*

Some time domain plots would help us see the fission.

(4) Nonlinear absorption and relation to reduced observed bandwidth

- Given that there is a strong non-instantaneous nonlinear refractive component in the CS₂, one would expect a complementary strong non-instantaneous nonlinear absorption. As an analogy, Kerr has the complementary two-photon absorption. What is the equivalent non-instantaneous absorption mechanism here?

Do the authors have a power in-power out curve? It is possible to extract the nonlinear absorption following the methodology in the literature. See for example, Aitchison et al, IEEE JQE 1997. Of course the absorption might have a different scaling and equation.

This could perhaps help explain the difference between experiment and theory in terms of spectral broadening.

- This also fits in with the discussion on the bottom of page 5 and top of page 8.

In particular, the section where you do the simulations on this point, pure Kerr, pure non-instantaneous, and mixed is one possible explanation. However, pure-Kerr with two-photon absorption also has restricted bandwidth. See for example: Yin, Agrawal (Opt. Lett. 2007) and Hsieh, Osgood (Opt. Exp. 2007)

- GVD could also impact the broadening width. How do we know the spectral width is not affected by an uncertainty in the GVD? Have you simulated a small change to the GVD to see how much this changes?

(5) Relation to reference 42.

Reference 42 operates in the AD regime inside a non-instantaneous liquid fiber similar to here. They also show supercontinuum generation and claim the origin is soliton fission. While it is true that Ref. 42 does not analyze linearon dynamics, a similar claim could be made here in that the majority of evidence for linearons is in the form numerical simulations, and therefore not a sufficient advance compared to what is known. Notably, in the introduction, the AD is claimed as a novelty, but it was already shown in this earlier work. This reference should be featured in the introductory material or at least described in further detail as it very much presents similar work to this manuscript.

(6) Paper length

The paper is long and some sections could be moved to supporting material. In particular I suggest moving:

- the second paragraph (p3) describing the dispersion model. While a helpful technical point, this is not the main focus of this paper and a distraction from the main point of the paper.
- The end section starting with 'Influence of uncertainties...'
- The observation "Second, we observe fine spectral fringes" is also a good point as it contrasts with the noisy Kerr.

Maybe focus on the NSR observation and its analysis in a first paper. If a coherence measurement is possible, report that as a separate paper. This would help reduce the length of the manuscript and keep it focused.

TECHNICAL POINTS (MINOR)

Good point to add the data showing the mode maintains single-mode nature.

In the paper there are peaks in the experimental spectrum presented as evidence of linearon dynamics. This seems pretty reasonable and could probably be a good point of evidence. However, there is no direct comparison of experiment with simulation in cross-cut as in Fig. 3(a). A separate figure is okay as these highly nonlinear systems rarely exhibit overlap between theory and experiment.

“In this work, we present an indirect approach to reveal linearons and their dynamics utilizing soliton-based supercontinuum generation (SCG) in the anomalous dispersion regime,”

It's not unreasonable, but how do we know that linearons exhibit fission? It would be helpful to describe to what extent linearons do and do not share properties with conventional solitons.

Is there any inhomogeneity of the liquid along the fiber?

What is the pulse spectral bandwidth? Are the pulses transform limited?

Figure 1(a) is called out of order in the text. Also, the different effects in the figure are not described. It would be helpful to describe them in the text, supplement, or minimally call a reference describing them.

P6 - Soliton shower. The soliton shower as stated by Biancalana is a very specific concept. I would be careful about using that word here unless the authors are sure it applies to the same idea. I recall it having to do with modulation instability (Plasma-induced asymmetric self-phase modulation and modulational instability in gas-filled hollow-core photonic crystal fibers, PRL 2012)

Figure 4.

Which direction is delay or advance in time axis?

Steep edges are mentioned, however they are not obvious from these shapes.

Ref. 23 comments on bubble formation at 46C at ambient pressure. What is the pressure and temperature here? Is there any evidence of bubble formation? Literature or measurements showing stability would help.

CLARIFICATIONS/REFERENCES

“...in case of high soliton numbers ($N \ll 10$)”

Do you mean high or low soliton number? This comment is less important given the earlier statement on the role of soliton number in this paper.

What does ‘neat’ mean for CS2? This is in other papers but perhaps not clear to the nonlinear optics community.

P5 top. Which of these nonlinearities are you using here?

Is n_2 , m_2 and n_2 , e_1 known at all the wavelengths used here? Did you measure them or from a reference? If estimated that is fine, just state that.

This is repeated here: “Our GNSE avoids Taylor expansion of the propagation constant and includes absorption, dispersion and nonlinearity with their full spectral dependence across the entire wavelength region of interest.”

Methods: Modal attenuation of 14.5%. Does this mean the fraction lost or amount transmitted?

P5 – what is the typical energy step size of the experiment?

Is the pulse duration measured? What method?

Are there any free parameters in the simulations?

Fig. 6. Add the word ‘simulated’ or ‘numerical’ to make this clear.

GENERAL GRAMMAR CHECK AND WORDING

Here are a few, non-exhaustive examples.

P1, 3rd paragraph ‘dominating’ → dominant

“offering ideal non-instantaneous nonlinear properties for the excitation of linearons” → This field seems new. It would be prudent to reserve the word ‘ideal’ for consensus in the field and use something more along the lines of ‘promising’.

Same goes for p 9.

P5 bottom (and a few other locations) – check on using the word ‘extend’ vs. ‘extent’ (also p8)

In the following we answer in detail on the questions of the former reviewers. In the list below you will find the reviewers' comments in black and our responses in blue.

Reviewer 1:

The authors present an experimental study of supercontinuum generation in a fiber with a CS₂ liquid core. The size of the core has been carefully chosen to access the anomalous dispersion regime so that the continuum is governed by soliton dynamics. The supercontinuum that is generated covers a wavelength range of ~1-2.7 μ m, which does appear to be the largest continuum reported to date in this liquid (though not by a significant amount as Ref. [23] reports a continuum from 1.2-2.4 μ m). However, the main claim of this work is not the continuum itself, but the observation of a new kind of non-instantaneous soliton state that arises due to the delayed Raman response of the liquid.

The discovery of new pulse solutions is always of fundamental interest, and the authors make a case that such nonlocal soliton dynamics will allow for the generation of more coherent continuum sources, which would certainly be useful.

We are grateful to hear the Reviewer's opinion on our findings and greatly appreciate her/his feedback.

R1.1 However, my main concern with this paper is not related to the novelty of these solutions, but to how convincing their evidence is for their observation. From what I understand there are two distinguishing features of their spectrum that point to the excitation of the nonlocal solitons. The first is the reduced bandwidth of the continuum. On this point, how can they be sure that this isn't due to any additional losses in their system, or indeed any deviations in the dispersion properties of the fibers from their predictions (i.e., it is easy to image that movement and/or thermal gradients of the liquid core could result in variations in the refractive index)?

A1.1 The Reviewer is correct that we see indications for non-instantaneous soliton dynamics because of the overall reduced bandwidth and the distinct spectral features on the soliton side of the spectrum. First of all we want to point out, that linking spectral information to effects originating from the temporal response of the nonlinear medium is not a new methodology. It is well known and accepted in the community that a fast non-instantaneous response like Raman is responsible for pronounced red-shifts of optical solitons [Agrawal, Nonlinear Fiber Optics, Ed. 5, Academic Press, pp. 176, 2003]. Thus, bandwidth and distinct spectral features in the supercontinua can clearly be associated with the underlying nonlinear and soliton dynamics. Here we take advantage of this knowledge to deduce clear indicators for an altered soliton dynamics originating from the slow non-instantaneous response of molecular motions.

The concern about unknown additional losses in our system is reasonable, but can be resolved by further looking on the evolution of the long wavelength edge of the measured output spectra. Additional loss terms could only arise from waveguide loss due to (1) critical guidance (i.e. micro-bends or surface scattering) arising at V-parameter below unity, (2) impurities like water contamination introducing broadband loss towards the mid-IR due to the OH-vibrational band, or (3) nonlinear absorption. Critical guidance can be excluded since our fiber guides the fundamental mode at a V-parameter >3 over the entire bandwidth (see Fig. R1). As a practical example for the quality of the liquid-core fibers, we measured a high transmission efficiency of above 15% behind

after a 20 cm long CCl_4 -filled capillary, which had a very low V-parameter of the fundamental mode of only 1.2.

Fig. R1: Spectral distribution of the V-parameter of the CS_2 /silica fiber (red) and the supported fundamental mode (blue).

Secondly, additional absorption bands would become visible in the output spectra by an intensity decrease of the long-wavelength edge for a further shift into the mid-IR. However, the long wavelength edge remains constant in intensity as it shifts further into the mid-IR. In addition, the input/output power characteristic of our system (see Fig. R4 in A2.5) correlates well with our absorption model of the simulations. No indications of additional loss bands on the long wavelength side are visible in the data.

Finally, we show in answer A2.5 that nonlinear absorption is not observable in our data, too.

The uncertainties originating from dispersion inaccuracies we discussed in the original and the recent version of the manuscript in the section "Impact of uncertainties of pulse dispersion and nonlinear response". We compared our dispersion model with two other dispersion models ([Kedenburg et al, *Opt. Mater. Express* 2 (11), 1588, 2012], [Samoc, *J. Appl. Phys.* 94 (9), 6167, 2003]). All models yield a significantly reduced bandwidth of the hybrid systems compared to the instantaneous (classical) broadening and show similar spectral characteristics for increasing input power. However, our model results in a spectral bandwidth and a supercontinuum onset energy closest to the measurements, and therefore we believe that it is most accurate, particular from the physics perspective as we include the long-wavelength resonance of CS_2 . We did not include dispersion variations due to thermal load or liquid flow. However, due to the good match between the long wavelength edge of the measured spectra and the soliton wavelength calculated from the phase matching condition (see Fig. R3 in A2.4c) we are convinced that those effects can be neglected.

R1.2 The second appears to be based on the fine spectral features seen in Fig. 3(a) (identified by the diamonds). However, this spectrum is not compared with the simulations or anything else that could confirm the appearance of the solitons, and so there seems to be a lack of supporting evidence for this claim. Thus before this paper is accepted it would be helpful to see some stronger evidence presented for the observation of these non-instantaneous soliton solutions.

A1.2 We thank the Reviewer for his remarks. Indeed, we argue that the appearance of clear visibility of spectral features in the averaged spectra at soliton numbers as high as 100 and more are a clear

indication for an improved pulse-to-pulse coherence or a high degree of reproducibility of the spectra.

In the recent manuscript we put the focus on a more concise explanation of this statement. In simulations we show that, e.g., for pump pulses as long as 500 fs modulation instabilities dominate the spectral broadening in an instantaneous system characterized by distinct spectral side lobes emerging before the supercontinuum onset, yielding in very smooth spectral envelopes of the average spectra. It is widely accepted in the community that smooth spectra result from chaotic soliton dynamics and low pulse-to-pulse coherence.

The measured spectra from the liquid-core fibers, however, show distinct spectral fringes with modulation contrast of approx. 10 dB and a significantly reduced bandwidth compared to glass fibers with only instantaneous response. As requested by the Reviewer, we directly compare one of the measured spectra with a simulated average spectrum (100 runs with initial phase noise) in Fig. 6 in the recent manuscript, both showing very similar spectral fringes. From our point of view there is no other explanation for the visibility of those fringes at such high soliton numbers.

To further support this statement we measured the supercontinuum from an indium fluoride fiber with 9 μm core diameter pumped with a similar system at 1.92 μm (450 fs). We compare the output spectra for two different soliton numbers in Fig. R2. We see that the spectrum with the low soliton number carries much more distinct spectral features on the soliton side with a modulation contrast of approx. 10 dB than the spectrum with the high soliton number. The spectrum for $N = 22$ is much flatter on the soliton side and even spectral features like the spectral shape of the first Raman-shifted soliton vanish.

Fig. R2. Measured spectral distribution of supercontinua generated in 2 m long InF_3 -fiber with a core diameter of 9 μm (ZDW: 1.75 μm) pumped with a 450 fs pulse at 1.92 μm at two input power levels, i.e. soliton numbers (blue: $N = 16$, light green: $N = 22$).

As explained in the manuscript, we use the simulations to predict that a drastic increase of the spectral pulse-to-pulse coherence (due to the non-instantaneous response of the liquid) is responsible for the observed spectral features on the soliton side. This is a completely unknown and new effect which can be generalized to all media with a highly non-instantaneous nonlinearity featuring a response time much longer than the initial optical pulse. We discuss the experimental issues of a direct experimental proof of the increased inter-pulse coherence below in A1.11.

However, we clearly state in the manuscript that the improved spectral coherence remains a prediction which we see by the observation of the distinct spectral fringes, overall representing an indication but not a proof. An additional coherence measurement is, from our perspective, not required since we identify three further indications of the modified soliton dynamics: the reduced spectral bandwidth, the increased supercontinuum onset energy, and a hybrid phase matching

condition well correlating the dispersive wave side with the soliton side. Together with multiple simulations confirming good knowledge about the investigated fiber system we think the evidences unambiguously confirm the predicted hybrid soliton dynamics.

I also have a few other comments and suggestions for the authors, as outlined below.

R1.3 The authors refer to these non-instantaneous soliton solutions as "linearons". From what I can tell this is the first published use of this term and thus perhaps some explanation is required. I would say that I am not a great fan of this term myself, but it is up to the authors to decide if they want to introduce it to the community.

A1.3 We understand the Reviewer's concerns about introducing a completely new and also uncertain terminology. The term "linearon" was firstly used by Conti and coworker as a working title for their non-instantaneous solitons since it is a solution of a quasi-linear propagation equation. We understand that this term could cause confusions. Furthermore, in our system it is not possible to excite a clean non-instantaneous state but rather an intermediate or hybrid state between a classical Kerr soliton and a non-instantaneous soliton. Therefore, we follow the Reviewer's recommendation and changed our terminology from "linearon" to "hybrid solitary wave (HSW)" throughout the manuscript.

R1.4 In my opinion Fig. 1(c) is too small for the amount of information it contains. Also, the authors talk about working close to the first zero dispersion wavelength (ZDW), however, I can only see one ZDW in the dispersion plot of Fig. 1(c) - not two. They need to explain this better, and expand the plot if necessary.

A1.4 We agree with the Reviewer. We tried to trim the figures to be easily processable for a two column article format. However, since this is a typesetting issue for later we changed the figure format in the recent manuscript. The corresponding figure (now Fig. 4) was updated and increased in size. We furthermore do not use the term "first ZDW" anymore since we only discuss the first ZDW.

R1.5 I have no idea what message they are trying to convey with the statement below: "From the experimental point of view, domain (i) is critical in terms of coupling, guidance and fiber nonlinearity due to a weak field confinement within the nonlinear core."

A1.5 Dispersion domain (i) is very common for micro-core glass fibers or tapered fibers. For decreasing core diameters the fundamental mode is less confined leading to a large extent of the field into the cladding. As a consequence, the waveguide dispersion converges to the material dispersion of the cladding. Even for a long zero dispersion wavelength (ZDW) of the core material shifting the lower ZDW to experimentally accessible wavelengths requires sub-wavelength core sizes. This effect is used for dispersion engineering in the waveguide structure such as soft-glass step-index fibers. However, those fiber structures are not very practical since coupling, wave guiding, and nonlinear gain are critical due to the low V-parameter and the very low coupling efficiency.

However, to comply with the Reviewer's opinion and since this is not an essential information to understand the argumentation on the manuscript, we spare out this information and talk instead

only about the second local minimum of the ZDW at larger core diameters, which is a unique feature in liquid-core fibers (see Fig. 4 in the recent manuscript). We changed the paragraph accordingly.

R1.6 The authors use wavelength dependent values for their material parameters, however, the values for n_2 in Ref. [13] only cover from 500-1500nm. How do they estimate the values beyond this? There seems to be much emphasis on the model for the linear refractive index, and not so much on n_2 .

A1.6 We thank the Reviewer for checking our simulation model in detail. In fact, the model for the nonlinear refractive index is equally important as that used for the linear index in case of broadband light generation. The model for the nonlinear refractive index n_2 is in fact not part of our achievements and therefore we just rephrase the essential parts which are directly taken from [Reichert et al., Optica 1 (6), 436, 2014] (Ref. [16] in the recent manuscript) including the wavelength dependence of the electronic ($n_{2,el}$) and molecular contribution ($n_{2,mol}$) of n_2 . The electronic contribution is calculated from a third order susceptibility $\chi^{(3)}$, which is assumed to be wavelength independent due to the vanishing nonlinear absorption cross section in the near-IR. Due to much weaker resonances of CS₂ in the mid-IR than in the UV, we treat $\chi^{(3)}$ as wavelength independent in the mid-IR. However, $n_{2,el} = 3\chi^{(3)}(4n(\omega)^2\epsilon_0c_0)^{-1}$ follows the inverse wavelength dependence of the linear refractive index. To meet the Reviewer's comment, we added the equation for $n_{2,el}$ to the Materials & Methods section.

The wavelength dependency of the molecular contribution is taken into account by a spectral overlap integral according to Reichert et al. Due to the complexity of this model, we would like to refer the Reader to the cited literature at this point.

R1.7. It is not clear to me what causes the drop in power due to high peak powers. For high average powers I understand that there is damage to the material, but I can't see an explanation for the case of high peak powers. Is the drop in power reversible or is this also due to some permanent change in the system?

A1.7 To understand the origin of this type of damage is important to enable the use of low repetition rate pulsed laser systems like Ti:Sapphire pumped optical parametric amplifiers (OPA) to generate supercontinua in the liquid-core fibers. We investigated the origins of the fiber damage carefully but, however, could not find a definite reason for the peak power damage up to now.

We understand the average power induced damage as a result of an accumulated thermal load in the liquids that causes dissociation of CS₂ visible by the yellow debris at the fiber input (see Fig. A2c in the Suppl. Mat.). In case of peak power damage we anticipate a different origin since no debris was visible in such damaged fiber samples and the capillary channel looked clean under a microscope. We look for an irreversible process caused by a single pulse propagating through the fiber. We estimated that the energy of a single pulse per mole CS₂ is three orders of magnitude lower than the dissociation energy of CS₂. However, self-focusing could lead to a strong local field enhancement and the peak intensity might approach the single-pulse dissociation limit. Another effect could be void or bubble formation within fiber due to the high field intensities as reported earlier by Kedenburg et al and Churin et al (Ref. [32] and [33] in the recent manuscript) which, however, was never directly observed in our experiments.

We will further look for a reason of the peak power induced-damage in the bulk liquid using high peak power femtosecond pulses to monitor the ongoing processes.

R1.8 Fig. 3 compares the experimentally generated supercontinuum with their numerical model, generally showing a reasonable agreement. However, it would be helpful to show a direct comparison between the experiments and the model for one input energy to back up their statements about the appearance of the "fine features" that define the soliton behavior (as discussed on page 8). This could simply be achieved by overlapping the simulated spectrum in the top part of (a).

A1.8a We thank the Reviewer for this remark. We followed his recommendation and compare a single measured spectrum with a simulated average spectrum in the recent manuscript in Fig. 6a which shows a remarkable match in bandwidth and distinct spectral features. Furthermore, we highlighted a few spectral fringes and the 20 dB bandwidth to make it easier for the reader to follow the argumentation.

On this note, can the authors comment on why such distinguishing spectral features have not been seen in the previous continuums generated in this material, which have also been pumped with femtosecond pulses?

A1.8b To our knowledge there is only one experimental demonstration where the authors came close to the anomalous dispersion regime of this material (Ref. 33 in the current manuscript: Kedenburg et al., Opt Express 23 (7), 2015). The authors of this work pumped a CS₂-filled capillary with 5 μm and 10 μm core diameter clearly in the normal dispersion domain (see Fig. 4 in the recent manuscript) and the red edge of the self-phase modulation broadened spectrum eventually entered the anomalous dispersion domain. Only the 5 μm capillary shows significant broadening. In the case of a pumping wavelength at 1560 nm, the part of the spectrum which entered the anomalous dispersion domain (above 1.8 μm) shows similar spectral features as we report even though less distinct and with a questionable origin. In case of pumping at 1685 nm wavelength, the spectrum looks rather smooth, unlike in our measurements. This can have different reasons, like the spectral smoothing by other nonlinear effects (e.g. FWM, XPM) close to the zero dispersion wavelength, instabilities in peak power or pulse duration of the pump source, or overlaying supercontinuum spectra generated by higher order modes. Given the leak of information and data (only one spectrum per pump wavelength and fiber diameter, no mode pictures), and the operation in another dispersion regime, this question can unfortunately not be answered by us

However, we can exclude all these sources of uncertainties in our work. We clearly pump a fundamental mode in the anomalous dispersion regime with a highly stabilized laser system and we can identify all dominating effects involved in the process.

Related to this point, can the authors also show an evolution plot for the propagation in the temporal domain obtained from the model? Their previous paper (Ref. [9]), where these solitary pulses were proposed, largely focuses on the temporal features of these pulses and thus it would be good to know if any distinguishing features can be seen in this domain.

A1.8c We thank the Reviewer for this remark. In response to this request we entirely restructured our paper and severely improved the theoretical introduction to our work. We carefully explain the temporal and spectral features of the solitary states that we can expect in classical (instantaneous), entirely non-instantaneous, and hybrid systems including both types of nonlinear response (see Fig. 1). In the non-instantaneous case, the pulse characteristics is very similar to the solitons found in the work of Conti et al. We then go on to a concise explanation of the spectral and temporal features we can expect from supercontinua generated in the respective systems (Fig. 2). Here, we also show that the tempo-spectral signature of hybrid solitary waves can be found in the simulated supercontinua. We hope this satisfies the Reviewers request and gives the reader a much better insight how our work relates to the previous one.

R1.9. In Fig. 3(a), I am not sure that the mode profiles are enough to claim single mode excitation. Have they consider S^2 measurements and/or at least fitting the expected profiles with the fundamental mode? How many modes would they expect this fiber to support given the core size and the index of CS₂? Some further comment to back up this statement would help.

A1.9 We understand the concerns of the Reviewer. Our CS₂ fiber is a few-mode fiber with a V-parameter of 5 at 2 μm wavelength which is considerably above the single mode criterion ($V = 2.405$). Thus, it supports up to 7 modes, which may lower the energy coupling to the fundamental mode (HE₁₁) and thus the field content propagating in the anomalous dispersion regime. However, the modes are clearly distinguishable in their effective refractive index, and thus their numerical aperture. Furthermore, only two modes (HE₁₁, HE₁₃) feature intensity maxima in the center of the core, leading to a large mode overlap with the Gaussian input mode. The HE₁₃ mode, however, experiences higher losses due to microbends (i.e. much smaller V parameter than the fundamental mode), is highly normal dispersive, and thus would barely contribute to the broadening in our experiment. Due to the variation of the transverse field pattern the probability to excite this mode with a Gaussian beam is rather negligible, too.

The other modes are ring type-modes whose excitation is easily visible in the output mode pattern. To avoid coupling to those higher modes we took special considerations in our experiment. We used a high resolution IR camera to monitor the near-field image of the output mode. Higher order modes were clearly visible in case of inaccurate coupling and could be easily suppressed by adjusting the incident angle and the position of the input beam carefully. Also the beam diameter was optimized to the numerical aperture of the coupling system by a telescope such that the fundamental mode could be effectively excited for our experiment.

In general, mode pictures are not enough to confirm the single-mode operation. However, in our measurement there is a further indicator that proves efficient coupling to the fundamental mode, which is the supercontinuum onset energy. If a high fraction (e.g. >10%) of the field would be coupled into higher order modes (e.g. the HE₁₃), this energy would not contribute to the soliton-based broadening of the fundamental mode, but to a parasitic side process, and the supercontinuum onset energy of the fundamental mode would appear at significant higher pulse energies. Compared to our simulation, only involving the fundamental mode, this would result in a higher supercontinuum onset energy, which is not the case. Rather the onset energies of the measurement and the simulation match very well (see i.e. Fig. 6b in the recent manuscript). This is a clear indication that most of the energy is effectively coupled into the fundamental mode.

Nevertheless, we understand that this issue might appear to the reader. To avoid confusion and to comply with the Reviewer's comment we added more information about all supported modes of the investigated waveguide into the supplemental materials (dispersion, V-parameter, relative nonlinear parameter, intensity distributions), and we briefly explain why the fundamental HE_{11} mode is dominantly excited in our system.

R1.10 On page 8 they have a discussion about the non-solitonic radiation (blue side) being bound to the solitons on the red side of the spectrum. Again, it would be helpful to see some temporal plots which show evidence of soliton formation to directly map their appearance to the spectral features. This is fairly common in papers reporting soliton-induced supercontinuum.

A1.10 We agree with the Reviewer that this is fairly common in the literature and we changed this accordingly to answer A1.8c.

R1.11 On pages 11-12 the authors discuss the coherence properties of the generated continuum where they make the claim that the linearon-induced continuum is more coherent than a continuum generated from an instantaneous nonlinearity. This section would be strengthened if the authors could conduct the experiments as well. For example, coherence measurements have been performed over a wavelength range of 1500-1800nm in the paper: Leo et al. Opt. Exp. v.22., p.28997 (2014) and something similar could be done here.

A1.11 We understand the concerns of the Reviewer and we agree that a direct measurement of the improved coherence would strengthen our claim about the improved coherence. However, coherence measurements in the mid-infrared wavelength domain are so far unexplored and require a new experimental methodology which is beyond the scope of this manuscript, as we explain in more detail in the following. Instead, we provide four reliable experimental indicators which strongly link the spectral features of our measurement to a hybrid soliton dynamics. To our knowledge there are no other effects that could explain the spectral fingerprint on a comparable level of evidence as we do here by linking of observables to simulations which is a commonly accepted methodology in the supercontinuum generation community.

Nevertheless, we want to give an impression of the experimental challenges of coherence measurements on the short-wavelength side of the infrared spectrum. We list the common methods to measure the pulse-to-pulse stability of supercontinua in the following together with their technical challenges:

1. The Reviewer highlights a work in which the technique of unbalanced Michelson interferometry was used to measure the fringe visibility of two interfering individual supercontinua which correlates with the first degree of coherence. Here, one supercontinuum has to overlap temporally with a second supercontinuum generated by a second subsequent pulse.

In our experiment, this method is not applicable since we use a laser system with a maximum pulse repetition rate of 11 MHz. The optical path difference which is required to bridge the pulse rate is 27.3 m. An interferometer arm length of more than 13.6 m is necessary which simply exceeds our laboratory space. Even if a series of mirrors is used to fold the optical path, strong measurement errors have to be expected due to atmospheric

distortions of the optical beam. A laser system with a higher pulse repetition rate will enable such a measurement in the future.

2. A second method requires synchronously pumping of two identical fibers with a 50:50 split pulse and spatial overlapping of the individual output spectra in an optical spectral analyzer. The fringe visibility should correspond to the first degree of coherence.

For liquid-core fibers, it is yet unknown if small variations of the local liquid pressure, the temperature, induced liquid flow, or fiber length could cause drastic changes in the interferogram.

3. An indirect method is the measurement of single spectra by applying a strong homogenous chirp on the entire supercontinuum pulse train and using a ultrafast photodiode to detect the chirped pulse with a fast sampling scope. The spectral intensity is effectively mapped into the temporal domain, which is referred as dispersive fourier transform [Goda and Jalali, Nat. Photonics 7, 2013]. Although the spectral intensity cannot be used to calculate the first degree of coherence this method allows the measurement of thousands of single pulse spectra whose stability correlate with the mentioned coherence [Klimczak et al, Sci. Rep. 6, 19284, 2016].

This method works especially good at telecommunications wavelengths due to the availability of cost-effective kilometer long fibers with low loss and ultrafast photodiodes with few 100 GHz bandwidth. Both are not available for the short-wavelength side of the infrared wavelength region in which we work (e.g. 5m of ZnF₃ single-mode fiber have approx. 150 dB/km; Thorlabs InGaAs photodiode DET05: bandwidth < 60 GHz).

4. Within the previous method, the dispersive element (here long-fiber) can be replaced by a refractive element like a grating or a prism to spread the wavelength components spatially along a rail. The spatial distribution can be calibrated with a multimode fiber on that rail and a fast photodiode can be used to measure a pulse trace per wavelength window. The speed of the photodiode is irrelevant as long the diode signals do not overlap (e.g. bandwidth > 1000 x repetition rate). The diode current correlates with the photon number and thus the integral over a single signal peak correlates with the pulse energy in the specific wavelength window. Similar to the first degree of coherence, it should be possible to define a degree of correlation for a series of signal peak integral, which can also be correlated to simulations.

This method also works in the mid-IR wavelength region since it relaxes the demands on the read-out speed of InGaAs photodiodes. We are currently conducting measurements based on this method and preliminary results show a good agreement between the measured broadband degree of correlation and simulations. However, it requires an accurate calibration method, another reference system (e.g. a supercontinuum from a glass fiber with comparable soliton number) and an analysis scheme which needs an extensive introduction.

Our conclusion is that we need a customized method here since our system operates at wavelengths and repetition rate which does not allow the straightforward use of one of the standard methods to measure the coherence. Here we believe that such a measurement (e.g. method 4) would exceed the scope of this paper, in fact we think that this kind of study would define a second paper. However, we hope in the light of our four well-founded experimental indicators we are able to satisfy the request of the Reviewer.

R1.12 At the top of page 13 the authors discuss some numerical simulations conducted in fibers using different Sellmeier models, however, I cannot find where these results are reported. Given that these are used to support their claim that the two-term Sellmeier is critical it would be helpful to see these somewhere, even if in the supplementary information. Similarly, the results of the modelling with different molecular fractions in gamma could also be shown in the supplementary information.

A1.12 The Reviewer is correct that we put a lot of emphasis on our new dispersion model. To support our model we added an additional figure (see Fig. A2) in the supplemental materials which compares supercontinuum simulations (spectra over fiber length) including the three different dispersion models (Samoc, Kedenburg and ours). The results clearly show that our model yields spectra with both smallest bandwidth and lowest supercontinuum onset point, both matching best to the experiment. A second figure in the supplemental materials shows the output spectrum of our system for an increasing molecular fraction (see Fig. A3), which clearly shows a reduction of the bandwidth with increasing f_m .

R1.13 In the conclusion they claim that one of the advantages of their system is that they don't need to use complex PCFs to generate a flat SC, but then say that they could over problems with the absorption of the liquids by using PCFs. These statements seem to cancel each other out.

A1.13 We thank the Reviewer for his/her comment. These two statements are made in different contexts and the use of a photonic crystal fiber (PCF) strongly depends on the requirements of the application. Our study offers a new approach how to address the anomalous dispersion regime in easy-to-fabricate liquid-filled capillaries for nonlinear light generation. This is a much simpler approach compared to selectively filling PCFs and allows insights in new physical processes in liquids appearing in this interesting dispersion regime with a straightforward experimental approach. However, we are aware of the drawbacks of working at wavelengths above 1.9 μm , like higher material absorption, slow and wavelength limited detectors, and limited availability of laser sources with high pulse repetition rate. Further suppression of the zero-dispersion wavelength towards more user-friendly wavelengths (e.g. telecom C- or L-band) can overcome those drawbacks. Up to now, we do not see an alternative way to reach those wavelengths without using more sophisticated selectively-filled PCF structures. We think that this difference is clearly stated in the manuscript and we hope for the appreciation of the Reviewer at this point.

R1.14 There are a number of typos and grammatical errors in the manuscript and references. I would recommend that the authors proofread this carefully before resubmitting.

A1.14 We again thank the Reviewer for the careful read. We thoroughly checked the manuscript for mistakes and hope that the Reviewer is convinced by the quality of the current version of the manuscript.

R1.15 Some of the references seem a little redundant. There are 5 for the refractive index of CS₂, and several of the reports seem to overlap in values and wavelength.

A1.15 We thank the Reviewer for this hint. To reduce the error of our model fit we included as many data as possible which is the reason for the large number of references related to this topic. However, since the model fit is shown in the supplemental materials, we referenced the used measurements of the refractive index in the supplemental materials only. We checked the main manuscript for redundant references again and changed quite a few related to other points, too, to overall reduce the number of references.

Reviewer 2:

R2.1 Summary

The authors present a study of non-instantaneous nonlinear optical dynamics inside a fiber waveguide filled with liquid. The target is to show so-called 'linearons' as established in Conti PRL 2010. The authors clearly state that this work represents an 'indication' of linearons in lieu of direct observation, which is a reasonable approach given what is known about the relationship between classical solitons and supercontinuum. The topic certainly seems worth pursuing, especially given the ability to generate coherent broadband beams. It should be emphasized that these are sound experiments. The challenge of even having access to the light sources and technology required to do the experiment is commendable.

We thank the Reviewer for the positive assessment of our approach and scientific idea which is discussed in this work.

Unfortunately the experimental evidence as well as the accompanying analysis falls far below the standard expected for a high impact paper and cannot be recommended for publication in Nature Communications.

The technical description has a number of deep flaws and does not currently meet the standard for publication in lesser journals. In particular, the manuscript fails to properly analyze the experiments following the criteria and formalism outlined in Conti PRL. This is especially true of the analysis of a soliton number and non-soliton radiation, which are misleading as will be outlined below. The conventional soliton world does not apply here. That said, this topic is new and it is apparent that this analysis is challenging. The authors will be able to correct this part with some effort.

One of the major claims of novelty is the use of anomalous dispersion (AD). Importantly, reference 42 appears to lay the foundation for this work and is left to only a brief mention at the very end of this paper. Advance over prior work would need to show a significant advance in understanding.

The broadening analysis presented is a reasonable first approach. However, it could have a number of origins besides linearons, namely nonlinear absorption, uncertainty in the experimental dispersion, or even inhomogeneity of dispersion and nonlinearity along the waveguide. These different mechanisms should be considered. Coherence is another potential piece of evidence. However, no measurements demonstrating coherence are demonstrated here.

The paper will not significantly influence the field. A direct proof of linearons (or linearon supercontinuum) for example, a coherence measurement (in lieu of simulation), along with convincing analysis that the NSR could only come from linearon dynamics is required for a convincing demonstration. The analysis should also rule out other effects.

A2.1a General remarks:

We respect the Reviewer's opinion in terms of the requested improvements of the theoretical depth. The remarks inspired us to entirely restructure the manuscript and to thoroughly work over the details of our work. However, we politely disagree in her/his opinion that the paper is below the standards of Nature Communications or even lesser journals. From our perspective, the points mentioned by the Reviewer, although surely all reasonable and understandable, do not justify such a terminating judgment. As a field-opening paper ("non-instantaneous solitons in liquid core fibers"), this manuscript can clearly not answer all open questions regarding an entirely new soliton dynamics. Its purpose is to show first experimental evidence of a new soliton domain in liquids and to give a first fundamental understanding of the underlying soliton dynamics in highly non-instantaneous liquids in correspondence to the existing theory.

We understand the relevance of a coherence measurement for a further evidence of the suggested hybrid soliton dynamics. We explained in answer A1.11 that a coherence measurement in the mid-infrared is technically highly demanding and possibly requires an unexplored methodology which would not help the recent manuscript to become more convincing. Furthermore, we do not see this to be relevant for the cogency of our data based on the high amount of indicators which were all convincingly supported by our theoretical model. The weight of our claims is not on the coherence features of the supercontinua, we rather present the coherence as an anticipated feature of such system, which additionally increases the value of all previous work done in the field. We concisely link other observable like bandwidth, supercontinuum onset energy and spectral location of the dispersive wave to our theory, and to our best knowledge we do not know any other effect in the classical soliton theory which could cause the observations made in our experiments which we reproduced multiple times.

We also respectfully disagree in her/his opinion that this manuscript will not influence the field. We talked with many well-known people from the supercontinuum community like Dr. John Travers, Dr. Fabio Biancalana, Dr. Nicolas Joly, and Dr. Amir Abdolvand, who were impressed by the quality of the experiment and simulation and the agreement between both. We have shown small parts of the work on a international symposia (like FiO 2015 and CLEO 2016) always in front of a large audience, and important representatives of the field like Prof. Ole Bang contacted us to get more information about our work. We are convinced that our work will influence the field perceptibly and inspire other groups to join our research for example to investigate spectral coherence and hybrid soliton interactions.

Regarding the quality of our work, we use state-of-the-art dispersion models and pulse-propagation solvers for our theoretical analysis, and top-of-the-edge lasers and characterization tools for our experiments. We have very precise knowledge about our fiber system and gained deep understanding of the underlying physics in our system. The used methodology of deducing physical origins from clear spectral features is well-known from Raman-active supercontinuum generation and widely accepted in the community, as the Reviewer kindly mentioned in his first words of the summary. For the first time, we give a link to the novel theory of non-instantaneous solitons and an idea how it is connected to soliton fission in realistic highly non-instantaneous liquids. On top of that we demonstrate the first entirely soliton-based supercontinuum generation in a non-instantaneously nonlinear fiber and are able to calculate the observed spectral features with remarkably good match

with a generalized and a specialized pulse propagation equations. On the basis of those achievements, the improved quality of the manuscript, and in light of our answers outlined below we like to politely ask the Reviewer to reconsider her/his opinion and decision about our manuscript.

A2.1b More specific remarks:

The Reviewer criticizes a too weak reference to the work of Vieweg et al. (formerly Ref. [42], now Ref. [23]) and claims that our study represents only a minor progress compared to their experiment. We would like to express our appreciation for the work done by Vieweg et al., which paved the way for fiber-based optofluidic nonlinear light sources. We actually believe that in case our work is published in Nature Communication, the work by Vieweg et al. will gain a boost in recognition by scientist from various fields. Nevertheless, we believe that our work is substantially different from the early work by Vieweg et al. since they used carbon tetrachloride (CCl₄) which is widely known to be a liquid with dominant Raman contributions, but has no contribution from picosecond-long processes like molecular reorientation due to its tetragonal molecule structure [McMorrow et al., IEEE J. Quantum Electr., 24(2), 1988]. In this sense, this liquid is rather similar to glass systems and the soliton dynamics rather relies on commonly known processes like Raman-driven soliton self-frequency shifts. This is fundamentally different in the case of carbon disulfide which we have used for our experiments, which features a dominant impact of molecular reorientation with a response time of 1.6ps.

However, we understand, that we cannot claim that we demonstrate soliton-based supercontinuum generation in liquids for the first time in general. This was actually never our intention as we also explicitly acknowledged similar work of other groups in the introduction of our paper (e.g. “SCG in the AD regime was demonstrated in water-filled hollow-core photonic crystal fibers [21, 22] [...]”). We politely apologize for the ambiguity in the former points of novelty. Our main claim is the observation of new soliton dynamics since we are able to pump a *highly non-instantaneous* liquid-core fiber in the anomalous dispersion regime for the first time. We also agree that we have to put the reference to Vieweg and coworkers on the same level as the work of Bozolan et al., which we changed in the recent version of the manuscript.

The Reviewer further criticizes that the observed reduction in bandwidth and distinct spectral features are only weak indications due to the impact of variations in absorption, dispersion and nonlinearity on the measurable quantities. We politely disagree at this point. We are aware of the free parameters of our system and one central part of our analysis therefore investigates the impact of material dispersion and nonlinear response on the observables. We are convinced that our improved analysis, a new specialized model, and new experimental evidence significantly consolidated the plausibility of our manuscript, which now includes all the mentioned points.

TECHNICAL POINTS (MAJOR)

R2.2 Definitions for the linearon and use of conventional soliton definitions

- a) The current manuscript incorrectly applies the traditional definition of soliton number to the linearon situation. The traditional definition of the soliton number $N^2 = \gamma * P * T_0^2 / \beta_2$ comes from the Kerr nonlinearity with GVD in a dimensionless nonlinear Schrodinger equation. This is not the case with linearons and applying that definition to the situation here does not make sense. (p8). The definition of the linearon number is given as N_{script} in Conti PRL in which they explicitly state: “The constant N cannot be written explicitly, and is

found by requiring that the total soliton energy is E ." I understand that the manuscript attempts to lean on collective knowledge as a means to understand the system. However the application of these definitions is misleading and incorrect.

- b) Linearon shape is not a hyperbolic secant. Thus the traditional soliton definitions do not apply. Related to this, the definition of full-width at half-maximum is given for a pulse of hyperbolic secant form. However, this is not the shape of a linearon as described in Conti PRL.
- c) Side note: For a conventional soliton system N refers to the injected soliton number. This is in contrast to the comment on p.8: "refers to the number of solitons appearing after the fission point." All comments mentioning the soliton number should be re-visited.

Here the Reviewer mention in her/his comments that the manuscript lacks of a stronger link between the findings and the theory proposed by Conti et al. in their PRL paper [PRL 105, 263902, 2010]. We partially agree with that and would like to emphasize that inside the recent version of the manuscript we were to provide much stronger links by (1) the definition of hybrid soliton-like state as an intermediate state between classical Kerr solitons and non-instantaneous solitons, (2) the identification of hybrid solitons in soliton fission based supercontinua in simulations whose spectral features clearly correlate with the experiment, and (3) the successful qualitative description of the measured spectra with a specialized Schrödinger equation deduced from the theory by Conti et al. From our point of view our findings put our measurements on solid grounds and create an intuitive picture of the physical processes in non-instantaneously nonlinear systems.

However, our linking method we use here is different to the comment of the reviewer to correlate our nonlinear system directly to the presented solution of Conti et al. In the following we discuss the obstacles of the methodology proposed by the Reviewer by answering on her/his three main points.

- a) The Reviewer is indeed correct that another soliton number has to be applied in case of non-instantaneous solitons. However, as the Reviewer cited correctly, also the work by Conti et al. cannot define an explicit quantity here. Instead, they define another quantity which rather refers to the order m of the linear mode which can be found in the temporal potential formed by the nonlinear response. This mode order only bears the information that the potential maximally contains m modes, but there is no correlation to the number of created fundamental solitons after the fission of an energetically higher order soliton in such a distorted system. In fact, it is not entirely clear from their work whether non-instantaneous fission can occur at all. This is where our work sets in. The origin of this mode order m appears to be rather unrelated to practical quantities like the fraction between dispersive and nonlinear length as it in the case of the classical soliton number. Furthermore, the response model by Conti et al. is an ideal exponential function and the authors do not comments on how the linear solution might vary if a realistic response function is considered which features a rise time and multiple non-instantaneous response terms (like in case of CS₂: molecular reorientation, liberation, and dipole-dipole interaction). Consequently, it remains questionable whether such a quantity actually exists and whether it has a similarly practical meaning as the classical soliton number. The answer to this question is beyond the scope of this work.

It also remains an open question how the formulation of a new quantity based on the ideal system discussed by Conti et al might help to quantify our system better since we are not dealing with pure non-instantaneous solitons but with hybrid solitary waves. We apologize

for the misunderstandings in case this was not clearly formulated. In the new theory section of our manuscript we put a strong focus in making this differentiation. Thank you for pointing this out.

Still, our intention is to compare the new system with quantities which are well known from the classical soliton theory. Unlike the statement of the Reviewer this comparison is not misleading since we clearly stated in manuscript that the classical soliton number can only serve as an upper estimate: *“As a result of the new dynamics, the classical soliton number can only serve as upper estimate for the number of soliton-like states created after the fission point.”* A slightly changed form of this sentence is still included in the recent manuscript.

- b) We confirm that the Reviewer’s argumentation about the shape of the Linearon is correct, whereas, however, we do not understand why this argumentation is for the study presented here. The exact shape of the Eigenfunction of the realistic system will be rather different to the system investigated in the Conti-paper due to the different shape of the nonlinear response function (the realistic non-instantaneous response is not purely exponential). Even if this Eigenfunction would be known, it will not be an Eigenfunction of the realistic system since this will be an intermediate solitonic state with contributions from the instantaneous and non-instantaneous response as we describe in the recent version of the manuscript.

In any case we do not have access to the pulse shape in the experiment. Here, to explain the spectral features in the measured spectra, we successfully used a pulse shape reconstructed from the measurement. We do not see any meaningful way how to introduce the Eigenfunction of the system in our considerations, even if it would be known.

- c) The classical soliton number N is commonly known for lossless waveguides with purely quadratic dispersion and thus its use always refers to an idealistic system. Taking one step further, a N^{th} order soliton propagating in a lossless system distorted by higher order dispersion or Raman scattering will fall apart into up to N fundamental solitons as described by Dudley et al. [Rev. Mod. Phys. 78(4), 2006]. This is how we intended to introduce the soliton number in our manuscript: From our perspective using the classical soliton number refers to a hypothetical situation and is, precisely speaking, not applicable for realistic systems anyhow which involve loss, higher order dispersion and more. However, it is widely used and serves as an upper estimate in the community for the number of fundamental quasi-solitons involved in the supercontinuum process. As we explained earlier, we just use this convention to make our system comparable.

R2.3 Pure linearon and linear losses

One wonders why pure linearons are not possible here. The manuscript states: “However, the absorption of CS2 does not allow low-loss guidance along several meters of fiber thus preventing a direct proof of linearons.”

However, Reference 23 has experimental measurements showing the CS2 is low loss except at 2.2 um. Did the authors measure this value directly for their own? How do we reconcile this difference? Perhaps there is also a technical limitation to the liquid fiber length, in terms of homogeneity or related.

A2.3 In the quoted sentence we refer to the fiber length compared to the dispersive length (our pulse length is 460 fs), with the latter being 3.3 m. For 100 fs the dispersive length decreases to

15 cm. To observe the formation of the fundamental soliton, propagation lengths of a few dispersive lengths are necessary to distinguish the pulse shape at the output from normal dispersive propagation, which is critical in terms of material loss.

We measured the loss of the bulk CS₂ up to 2.15 μm using a 1m cuvette revealing an absorption of approx. 1 dB/cm at 1.95 μm. We included the data in Fig. 4 of the recent manuscript. The measurement matches our absorption model well, which, however, slightly overestimates the material loss in our simulation. The model also includes the absorption at about 2.22 μm as reported in the previous Ref. [16] (Ref. [32] in the recent manuscript). We assume that the Reviewer refers to this reference, since it is the only one that shows a transmission spectrum in this spectral domain. However, this transmission spectrum is measured using a 1 cm long cuvette filled with CS₂, which is a rather short length compared to feasible fiber lengths and compared to the necessary accuracy of loss measurements being relevant for optical fibers (i.e., to determine losses of the order of dB/m). Clearly the single or even few soliton operation regime requires fiber length which are not achievable at the working wavelength due to high material attenuation. A system with shorter pulses (e.g. 100 fs) could be one solution, but shorter pulse impose a decreasing molecular fraction (e.g. from 86% to 53%) and the non-instantaneous impact on the pulse shape becomes less dominant and may not be measurable. We assume only a combination of shorter pulses (200-300 fs), an operation wavelength for which the material absorption is low and sources with higher repetition rates (e.g. 1.55 μm) would give access to this regime.

R2.4 Non-solitonic radiation

- (a) The manuscript appears to use the classic NSR definition. However, the conditions for non-solitonic radiation are different for linearons than conventional solitons as outlined in Conti PRL 2010: “The energy-dependent part is an extra contribution to the resonant condition that is unique for highly noninstantaneous solitons, and allows us to tune the frequency position of the emitted radiation by adjusting the total input pulse energy.”

This should be quantitatively described in this paper as part of the proof of linearons. Given that it is different NSR, any time that it is mentioned in the text it should be highlighted as this is quite different to conventional knowledge.

- (b) P6 – “The spectral evolution is characteristic for soliton-driven SCG: After initial self-phase modulation (SPM) a sudden increase of the spectral bandwidth at 2.5nJ is observed with a distinct short- wavelength shoulder (around 1.25μm) appearing far apart from λ_p.”

COMMENT: This suddenly emergent blue shoulder is typically associated with NSR and arises because the soliton bandwidth (at this narrow temporal width) overlaps with the linear phase matching of the NSR and seeds it with energy. The description here leads us to think it is fission itself and should be clarified.

- (c) On the positive side, it should be possible for the authors to use their spectral results in Fig. 3 to compare this with calculations, and see if their observation of NSR matches that predicted for linearon dynamics. This would fit well at the top of page 8 where they say ‘we can show’. This could go in the supplement (or even a specialized paper) where you describe the specific forms of these terms.

- (d) “The prominent red shoulder of the SPM spectra, which is also visible in the measurements, denotes strong temporal self-compression (self-steepening [25]) of the optical pulse.”

Comment: Red shoulders are typically due to the Raman effect. Another possibility is asymmetric SPM from some perturbation or loss. Ref. 42. claims theirs in CCl₄ comes from

Raman, though one would need to re-visit their simulations to verify this is indeed true. Conti discusses the suppression of Raman under certain conditions. Does your system meet those conditions? If not, what is the explanation for the red-shoulder? Did you attempt a minimized simulation with just the linear and self-steepening to see if this red shoulder could come from that?

Strong temporal self-compression arises from higher-order soliton compression, as opposed to the self-steepening mentioned here. Separately, in Conti PRL, they explicitly state (p1,c2) that the shock term (i.e. self-steepening) is neglected. That said, the authors could prove them to be incorrect by showing simulations with and without this term, or showing an analytic reason.

- (e) “Here, the higher-order soliton, which in fact is the initial input pulse, breaks up into multiple fundamental solitons on the long wavelength side and sheds energy towards shorter wavelength via NSR [5, 25].”

Some time domain plots would help us see the fission.

A2.4 We thank the Reviewer his recommendations.

- (a) The Reviewer correctly stated that the phase matching condition for dispersive wave generation has an additional nonlinear phase term, which has to be considered. However, as we explained before, we do not observe clean dynamics of either instantaneous or non-instantaneous solitons, but instead of a hybrid state. The underlying phase matching condition to generate a dispersive wave will be considerably different from the two ideal (i.e., extreme) states. In the optimal scenario, the nonlinear phase of the hybrid state will be again a mixture of the two phase shifts of the instantaneous and the non-instantaneous part weighted by f_m .

Since the mixing of these nonlinear phases did not seem straight forward for us at the time we prepared the first version of the manuscript, we initially used only the linear phase matching condition to predict the initial wavelength of the first soliton which matched the measurements already very well. We changed this in the recent version of the manuscript accordingly to the advices of the Reviewer in R2.4c such that we included the nonlinear phase contribution. Please see answer A2.4c for more information.

- (b) The Reviewer is correct about the introduction sentence to the soliton fission process, thank you. We changed this sentence to the following in the recent manuscript:

“The instantaneous system (case i, Fig. 2a) shows conventional soliton fission: after initial self-phase modulation a burst of fundamental solitons is released at a fission length of approx. 2 cm. Here, the strongly compressed pulse breaks up into multiple fundamental solitons on the long wavelength side which shed energy towards shorter wavelengths via dispersive wave generation, overall leading to a bandwidth which is just limited by absorption.”

- (c) In case of classical soliton fission the nonlinear phase shift is $\varphi_{NL} = \gamma_0 P_s$ with the nonlinear parameter γ_0 at ω_0 and peak power of the first solitons $P_s = P_0(2N - 1)^2/N^2$ [Kodama, Hasegawa, IEEE Photonics Techn. Lett. QE-23, 1987]. In case of entirely non-instantaneous solitons it is $\varphi_{NL} = \gamma_0 \mathcal{E}_p R(\omega)$ with the pulse energy \mathcal{E}_p and the normalized response in the Fourier space $R(\omega)$. It is important to note that the phase shift of the non-instantaneous phase shift is small compared to the Kerr phase shift. For example, the frequency distance between dispersive wave and initial soliton is 144.7 THz at 7 nJ pulse energy if only the linear

phase is considered. If we add the non-instantaneous phase (case ii in the recent manuscript) we find the difference to be 145.2 THz, but 158.2 THz if only instantaneous phase (case i) is assumed. Analogously to our specialized Schrödinger equation we can define a hybrid nonlinear phase shift $\varphi_{NL} = (1 - f_m)\gamma_0 P_s + f_m\gamma_0 \mathcal{E}_p R(\omega)$ for the hybrid response system (case iii) which results in an intermediate frequency difference of 147.8 THz. Following the Reviewer's request we calculated the soliton wavelengths at the point where the dispersive wave is generated using all three phase matching conditions. Accordingly to the advice of the Reviewer we explain the different phase matching conditions in the Materials & Methods.

The calculated wavelength should fit best to the long wavelength side of the spectrum at the supercontinuum onset (at ca. 2.5 nJ). For higher pulse energies the generation of the dispersive wave happens somewhere in the middle of the fiber and the soliton will further shift towards longer wavelength until it reaches the fiber end. We changed former Fig. 3 (Fig. 5 in the recent manuscript) such that it shows all three solutions. Compared to the experimental data, the phase matching condition confirms again that entirely instantaneous Kerr nonlinearity (case i) cannot be the origin for the observed broadening. Instead the hybrid case (iii) results in the best fit to the experiment since it describes the spectral location of the most red-shifted spectral component most accurately at low input energies above the fission energy (2.5 nJ to 7 nJ). We thank the Reviewer for her/his advice which allowed us to identify another indicator for the hybrid soliton dynamics.

Fig. R3. Updated plot of the experimental data (now Fig. 5) including the initial soliton wavelengths (dotted red curves on the FTIR side) for increasing input energy calculated for all three nonlinear phase matching conditions: entirely instantaneous (case i), entirely non-instantaneous (case ii), and hybrid system (case iii).

- (d) The Reviewer is correct that dominant features appearing on the long wavelength side of the pump wavelength are normally assigned to Raman induced effects. However, we do not talk about new features suddenly appearing on this side, but about the more pronounced shoulder at an early stage of the broadening below 2.5 nJ pulse energy which is dominated by self-phase modulation. Accordingly to the book *Nonlinear Fiber Optics* by Prof. Govind Agrawal this is a clear sign for strong self-steepening: *“Self-steepening results from the intensity dependence of the group velocity. Its effects on SPM were first considered in liquid nonlinear media and later extended to optical fibers. Self-steepening leads to an asymmetry in the SPM-broadened spectra of ultrashort pulses.”*

Higher-order soliton compression is a result of self-steepening. We know that this effect was not included in the PRL paper by Conti et al. for simplicity since it also acts as a distortion on the system. We acknowledge the hint by the Reviewer to put a correction on the theory by Conti et al.; however, as we mentioned before, such an extensive theoretical study, without doubting its importance for future work, exceeds the scope of this paper.

Instead, since it is not a key message and to reduce the length of the paper, we deleted the sentence about self-steepening in the recent version of the manuscript.

- (e) We thank the Reviewer for his hint to include some time domain plots. Along with the substantial improvement of the theoretical introduction into our work we also included such plots in the recent version of the manuscript.

R2.5 Nonlinear absorption and relation to reduced observed bandwidth

- (a) Given that there is a strong non-instantaneous nonlinear refractive component in the CS2, one would expect a complementary strong non-instantaneous nonlinear absorption. As an analogy, Kerr has the complementary two-photon absorption. What is the equivalent non-instantaneous absorption mechanism here?

Do the authors have a power in-power out curve? It is possible to extract the nonlinear absorption following the methodology in the literature. See for example, Aitchison et al, IEEE JQE 1997. Of course the absorption might have a different scaling and equation. This could perhaps help explain the difference between experiment and theory in terms of spectral broadening.

- (b) This also fits in with the discussion on the bottom of page 5 and top of page 8. In particular, the section where you do the simulations on this point, pure Kerr, pure non-instantaneous, and mixed is one possible explanation. However, pure- Kerr with two-photon absorption also has restricted bandwidth. See for example: Yin, Agrawal (Opt. Lett. 2007) and Hsieh, Osgood (Opt. Exp. 2007)

- (c) GVD could also impact the broadening width. How do we know the spectral width is not affected by an uncertainty in the GVD? Have you simulated a small change to the GVD to see how much this changes?

A2.5 The Reviewer is correct that both nonlinear absorption and GVD might limit the spectral drastically. Starting with the nonlinear absorption, Reichert et al. investigated this quantity thoroughly and found that nonlinear absorption decreases strongly toward the near-IR (measured domain approx. 500 to 1500 nm). Since the next strong absorption occurs in the mid-IR at 6.6 μm it is fair to assume that the contribution of nonlinear absorption is negligible within our generated bandwidth domain.

We also checked our input power/ output power dependency of our measurement again and compared it to simulation (see Fig. R2b). In fact, the power characteristics of our system reveals two linear absorption regimes, one for low pulse energies before the supercontinuum covers the strong absorption at 2.22 μm [Plyler and Humphreys, Journal of research of the National Bureau of Standards, 39, 1947], and one for higher input energies being associated with a substantially broader bandwidth.

In the first low-power regime, the experimental data fit perfectly to the simulation data. In the second regime the slope difference of the power characteristics might result from a slight mismatch between our absorption model especially at 2.22 μm and the realistic material absorption (see also

Fig. 4c in the recent manuscript). However, the change of the slope towards the second regime is visible in the simulations, too. In any case, no clear saturation effect of the average pulse energy at the output is visible, thus we believe it is appropriate to neglect nonlinear absorption in our study. We understand that this is a key question for the reader and to avoid confusion we included figure R4b in the supplemental materials now.

Fig. R4: (a) Simulated spectra for increasing input energy with highlighted low and high bandwidth domains. The transition energy of the domains is defined by the pulse energy where the red-shifted maximum at the long wavelength edge reaches the absorption at approx. 2.22 μm (red line) (b) Comparison of the input/output characteristic of our system between experiment and simulation. The colored domains correspond to the domains defined in (a).

Regarding the GVD, we discussed the impact of the dispersion in the section “Impact of uncertainties of pulse dispersion and nonlinear response” which we moved to Materials & Methods in the recent manuscript. As a response to both Reviewers we also put simulation results for three slightly different core diameters and for each of the existing CS₂-material dispersion models (Samoc model, Kedenburg model, our own model) into the supplemental materials. The simulations show that dispersion slightly influences the fission point in the fiber and the output bandwidth, but not by a significant amount. They also reveal that our new dispersion model yields the smallest bandwidth and the best match to the supercontinuum onset, revealing its relevance. Overall, we want to emphasize that the selection of one dispersion model does not significantly influence the key message of our study since the numerical bandwidth comparison between an instant Kerr system and a hybrid system always involves the same dispersion function. The reduction of the bandwidth is always visible and remains to be associated with the non-instantaneous part of the nonlinear response.

R2.6 Relation to reference 42.

Reference 42 operates in the AD regime inside a non-instantaneous liquid fiber similar to here. They also show supercontinuum generation and claim the origin is soliton fission. While it is true that Ref. 42 does not analyze linearon dynamics, a similar claim could be made here in that the majority of evidence for linearons is in the form numerical simulations, and therefore not a sufficient advance compared to what is known. Notably, in the introduction, the AD is claimed as a novelty, but it was already shown in this earlier work. This reference should be featured in the introductory material or at least described in further detail as it very much presents similar work to this manuscript.

A2.6 We agree with the Reviewer on the point that the mentioned reference needs to be highlighted in our introduction as it indeed demonstrates AD operation of liquid-core fibers. We thank the Reviewer for pointing this out.

However, as we already explained in our answer A2.1, Vieweg et al. worked with CCl_4 which does not show strong non-instantaneous nonlinearity within the scope of their experimental conditions since, to our knowledge, the instantaneous response of CCl_4 dominates at the pulses width used (200 fs). Thus the hybrid dynamics we report on here are not identifiable from their work without providing more information of the system.

In more detail:

A detailed analysis of the nonlinear Kerr response of CCl_4 is given in [McMorrow et al., IEEE J. Quantum Electr., 24(2), 1988]: *“ CCl_4 possesses an isotropic polarizability, all relaxations associated with orientational motion (including librations) are absent. [...] The net response consists of a sum of four terms [...]: an instantaneous signal component [...], two resonantly driven vibrational terms [...], and finally, a small rapidly decaying component exhibiting an 200 fs time constant [...]”*

Also in the response model used in the simulations by Vieweg et al [Itoh et al, Jpn. J. Appl. Phys. 43, 6448, 2004] molecular reorientation does not play any role. The two vibrational Raman terms are fast oscillating terms which have an intrinsically smaller contribution to the total nonlinear response due to their fundamentally different physical origin: In case of reorientation (CS_2), the molecules physically rotate (i.e., induced anisotropy of the molecules), leading to comparably large nonlinear change of the refractive index. This is different in case of Raman scattering (CCl_4), which is an intramolecular effect and thus causing overall much smaller index changes than reorientation. As a consequence, CCl_4 possesses a much weaker and faster non-instantaneous response relative to the 200 fs pulse width used in the experiments by Vieweg et al.

Just taking into account the various time constants of CCl_4 , we expect that the spectra presented by Vieweg et al. follow a rather classical soliton fission without any measureable contribution from the non-instantaneous part, but including very well-known effects like Raman-driven soliton self-frequency shifts. The authors also do not report on any unexpected broadening behavior or unusual spectral features, presumably because the hybrid soliton dynamics is not observable in their system. From their few data it is in fact hard to compare the benchmarks of the Vieweg system with ours, since key parameters are not stated (e.g., classical soliton number and supercontinuum onset energy are missing).

Our study identifies and discusses key observables in the spectra in great detail and correlates them with the theory by Conti et al. In the new version of the manuscript we put a strong emphasize on a new theory part explaining this difference to previous work. We therefore see a strong progress in our work compared to pervious work by other groups, which makes our work unique and valuable for a publication in Nature Communications.

R2.7 Paper length

The paper is long and some sections could be moved to supporting material. In particular I suggest moving:

- the second paragraph (p3) describing the dispersion model. While a helpful technical point, this is not the main focus of this paper and a distraction from the main point of the paper.

- The end section starting with ‘Influence of uncertainties...’
- The observation “Second, we observe fine spectral fringes” is also a good point as it contrasts with the noisy Kerr.

Maybe focus on the NSR observation and its analysis in a first paper. If a coherence measurement is possible, report that as a separate paper. This would help reduce the length of the manuscript and keep it focused.

A2.7 We thank the Reviewer for his/her helpful recommendations. We agree that the manuscript was quite long and contained many technical details. In the recent manuscript we moved a substantial fraction of the technical information to the Materials & Methods section, namely the experimental setup, the dispersion model, and the discussion about the influence of dispersion and nonlinearity on the simulation results, as requested by the Reviewer.

However, we left a short analysis of the coherence in the manuscript since (1) it is very common in the supercontinuum community to give a prediction about the first order coherence of the generated spectra, and (2) it is necessary to understand that the power characteristics we measured identify soliton fission as the dominant broadening process and not modulation instabilities as it has to be expected from instantaneous glass systems. We also had to add a small part to introduce the hybrid solitary waves as an essential part of the underlying physics in our nonlinear system. Sparing out these simulations will reduce the number of indicators for the new hybrid soliton dynamics and lower the impact of our work. We hope for the comprehension of the Reviewer and the editor at this point. We are convinced that our recent manuscript only contains the most necessary information for a complete story line which can easily be followed by the reader and which keeps a clear focus.

TECHNICAL POINTS (MINOR)

Good point to add the data showing the mode maintains single-mode nature.

We thank the Reviewer for his/her opinion. We would like to refer the Reviewer to our response in A1.9, where we explained that the good match between measured and calculated supercontinuum onset energy is a further sign for single-mode operation. This statement is also included in the recent manuscript and we hope that the Reviewer accepts this as a solution for his/her request, too.

In the paper there are peaks in the experimental spectrum presented as evidence of linearon dynamics. This seems pretty reasonable and could probably a good point of evidence. However, there is no direct comparison of experiment with simulation in cross- cut as in Fig. 3(a). A separate figure is okay as these highly nonlinear systems rarely exhibit overlap between theory and experiment.

Thank you for that comment. We show a direct comparison of experiment and simulation in a separate figure in the new version of the manuscript (see Fig. 6a). The overlap between distinct spectral feature (i.e. location of the dispersive wave and the modulation contrast of the spectral fringes on the long wavelength side) in both spectra is convincing from our point of view, as additionally pointed out by the Reviewer.

“In this work, we present an indirect approach to reveal linearons and their dynamics utilizing soliton-based supercontinuum generation (SCG) in the anomalous dispersion regime,”

It's not unreasonable, but how do we know that linearons exhibit fission? It would be helpful to describe to what extent linearons do and do not share properties with conventional solitons.

Again, we thank the Reviewer for this thought. As we mentioned earlier, we took this argument very serious and to complete reconsider the structure of the manuscript. In fact we applied many of her/his recommendations including this one in the new version of the manuscript (see i.e. the discussion to Fig. 2).

Is there any inhomogeneity of the liquid along the fiber?

We are sure that this was not the case. Any kind of distortion like formation of bubbles or streams in the microfluidic holders we immediately recognized in both the mode image on the camera and the output spectrum, which both turned out to be stable throughout the measurements.

What is the pulse spectral bandwidth? Are the pulses transform limited?

The pulses are not fully transform limited, but carry only a weak third-order chirp since they are compressed by a grating compressor. As we described in the manuscript, we reconstruct the pulses from their spectrum and auto-correlation using a script which iteratively adds a third-order phase on the Fourier-transformed spectrum until the numerical auto-correlation matches the measured one. The sign of the third-order chirp can be estimated from the materials used in the setup before compression. This estimation (with $\beta_3 = -0.025 \text{ ps}^3$) corresponded very well with simulations where we compared the fission energy of a negatively and a positively chirped pulse with the experiment.

Figure 1(a) is called out of order in the text. Also, the different effects in the figure are not described. It would be helpful to describe them in the text, supplement, or minimally call a reference describing them.

We significantly changed Fig. 1 in the recent version of the manuscript. We also moved parts of Fig. 1 to Fig. 4. With that we hope that we have sorted out the mentioned issue.

P6 - Soliton shower. The soliton shower as stated by Biancalana is a very specific concept. I would be careful about using that word here unless the authors are sure it applies to the same idea. I recall it having to do with modulation instability (Plasma- induced asymmetric self-phase modulation and modulational instability in gas-filled hollow-core photonic crystal fibers, PRL 2012)

We thank the Reviewer for this comment. We changed this term to "a burst of fundamental solitons".

Figure 4.

Which direction is delay or advance in time axis?

Steep edges are mentioned, however they are not obvious from these shapes.

The Reviewer is correct and we included adjectives like "trailing edges" to make clear which part of the pulse is leading and which is trailing.

Ref. 23 comments on bubble formation at 46C at ambient pressure. What is the pressure and temperature here? Is there any evidence of bubble formation? Literature or measurements showing stability would help.

We never directly observed bubble formation in our experiments. As we described in the paper we found power regimes where the fiber transmission dropped significantly, but we cannot confirm that this originate from bubble formation. We assume that bubble formation due to nonlinear absorption is not as dominant for our operation wavelength as for the wavelengths used in work by Kedenburg et al. since this kind of absorption is expected to become weaker towards the mid-infrared. Furthermore, our system operates at a much lower repetition rate, which reduced the average power and thus the thermal load on the liquid core material.

We take the opportunity to thank the Reviewers once again for their constructive comments and suggestions, which have helped us to substantially improve the quality of our manuscript. Having responded to their queries in detail, we look forward to your response.

Reviewers' comments:

Reviewer #1 (Remarks to the Author):

The authors have significantly modified their discussions related to the pulse dynamics in this paper. Most notably, they now acknowledge that their solutions are hybrid solitary waves that form due to the influence of both the instantaneous and non-instantaneous nonlinearities. Whilst I am not an expert in nonlocal soliton dynamics, in my view this looks to provide a much more accurate description of their results.

Overall I am happy with the revisions that the authors have made to this manuscript and their efforts to address my comments. Although there are still some answered questions, given the challenging nature of these experiments, they have done a sufficient job to convince me of their observations. I only have a few minor corrections below.

1. Fig. 4b is not called out in the main body of the text. Also, the description of Fig 4c is somewhat lacking. Can the authors please comment on the differences between the model absorption and that of the bulk measurement? I.e., how do they get from one to the other?
2. Similarly, Fig. 5b is not called out in the text and the discussion relating to 5c comes after Fig. 6. This makes it very difficult to understand these simulations when you first encounter them. Also, although I can see slight differences in these two spectrograms, it is not clear why the hybrid case is considered better than the GNLSE. The authors need to better explain why the hybrid Schrodinger equation is preferred over the GNLSE, else it is not clear why you would need to bother with this.
3. A minor point, the authors use the abbreviation GNSE for the generalized nonlinear Schrodinger equation. GNLSE is more common.
4. Although the text is perfectly understandable, there are still several typos, missing articles, plurals that should be singular etc. Also, when they refer to the pulses being hundreds of femtoseconds – they should mention they are talking about durations here. I would recommend they ask a native English speaker to proofread this, or ask one of the highly experienced co-authors to review it properly.

Reviewer #2 (Remarks to the Author):

Review of “Hybrid soliton dynamics in liquid-core fibers”

SUMMARY

This paper is greatly improved from the first version. The novelty of the research is much easier to understand (coherent soliton behavior). Importantly, the distinction between well-known solitons and the current work is now much clearer. This also goes for the recent theory work by Conti and its relative degree of applicability to the present case. The shift away from the discussion of ‘linearons’ and related was also echoed by the other reviewer. The more general ‘non-instantaneous nonlinearities’ is appreciated.

The revised presentation order is definitely helpful as it first lays out the theory and a physical description of the expected phenomenon before moving onto experiments. Apologies if the first review appeared harsh. While the high technical quality was noted, as written, I feared that the work would be lost on the vast majority of readers. I appreciate the authors took the advice of the reviewers into consideration to improve their work. It appears that the other reviewer suggested similar updates. This work represents a nice piece of science and the authors should keep up the good work.

There remain a few points to address before the work is ready for the public eye.

TECHNICAL COMMENTS

Coherence of the system

Two major points here.

1. Unity coherence?

Is this indeed fully coherent in Figure 3? While it is obvious that this system is more coherent than regular soliton supercontinuum, $g(1) = 100\%$ would be remarkable and is very surprising. Indeed, one could say it is too good to be true. If this is true, what are the conditions (molecular fraction, dispersion, other) to achieve this? There must be some practical limitations to the system. See the next point.

2. Last version versus this version.

This is all the more shocking given the striking difference between the earlier paper and this one. Figure 3a(left) replaces old Fig. 6a (right). The coherence of these two is completely different. Is all of this simply due to the change of 81 to 85% molecular fraction?

It is recommended the authors outline the sensitivity of this claim especially given the strong claim of full coherence. Thank you for pointing out the ‘stiffen’ mechanism for the phase noise in your paper.

Small points

- Page 4 – The new figure 1 is helpful to understand your system. Thank you.
- Figure 5.
What do the (i), (ii), (iii) correspond to? Please define in the text or caption.
- Nonlinear loss
It would be helpful to reference the analysis on nonlinear loss in the supplement in main paper, so it's clear to others this is considered.
Also note Supplement Fig 6, the legend should have a line instead of dots for the theory.
- RI?
This is not defined, but we can guess what it is. Do we really need to abbreviate refractive index?
- Definition of gamma
The standard nonlinear parameter is: $\gamma = k_0 n_2 / A_{\text{eff}}$. In the Methods section, the authors have a $A_{\text{eff}}^{(1/4)}$. Is this a different gamma? The units don't work out at present. This section needs to be re-visited.
- Reply letter – page 20
“Higher-order soliton compression is a result of self-steepening.”
Higher-order soliton compression results simply from higher-order solitons and does not require self-steepening to occur.

GENERAL GRAMMAR CHECK

This was mentioned in the first review. It appears these comments were not taken seriously. There are easily well over ten misspellings, missing words, and related grammar faults in the current version. I do not list them here as that's not the role of the reviewer. While the content comes through, these are a distraction to the reader and could result in them losing their focus. Beyond a careful look themselves, the authors should have some people outside their group read the paper.

Response letter to our manuscript NCOMMS-16-04149B

Thank you for providing further feedback to our revised manuscript. We are grateful to the Reviewers for the positive feedback and for the important points which have helped us to correct and improve the manuscript. In this letter, we respond on each comment of the two Reviewers and on the formatting issues. Please find our answers in blue and the changes in the recent manuscript in *italic red*.

Reviewer 1

The authors have significantly modified their discussions related to the pulse dynamics in this paper. Most notably, they now acknowledge that their solutions are hybrid solitary waves that form due to the influence of both the instantaneous and non-instantaneous nonlinearities. Whilst I am not an expert in nonlocal soliton dynamics, in my view this looks to provide a much more accurate description of their results.

Overall I am happy with the revisions that the authors have made to this manuscript and their efforts to address my comments. Although there are still some answered questions, given the challenging nature of these experiments, they have done a sufficient job to convince me of their observations. I only have a few minor corrections below.

We thank the Reviewer for the positive comments about the revised manuscript. We highly appreciate her/his feedback, which have allowed us to greatly improve the quality of the manuscript.

R1.1 Fig. 4b is not called out in the main body of the text. Also, the description of Fig 4c is somewhat lacking. Can the authors please comment on the differences between the model absorption and that of the bulk measurement? I.e., how do they get from one to the other?

We have corrected the manuscript and now call-out Fig. 4b at the location where we introduce the waveguide geometry which we used in the experiment. To account for the second comment of the Reviewer, we improved the paragraph in which we explain the fiber attenuation model (main text close to Fig. 4, ll. 260):

The modal attenuation of our system is governed by the material absorption of CS₂ (Fig. 4e), which was estimated on basis of in-house measurements and previously reported data (see Material & Methods). The absorption of the liquid is approximately four orders of magnitude larger than that of silica at 2 μm wavelength. However, guidance along few tens of centimeters inside a CS₂/silica fiber is possible with transmission values well above 30 % at this wavelength.

We furthermore added a detailed description of how we included the absorption into simulations to the section *Generalized Schrödinger equation* in Materials & Methods (II. 456):

We measured the absorption of CS₂ at near-infrared and visible wavelengths using a 1m-long metal tube (diameter 12mm) with plain sapphire windows positioned in the collimated probe beam of a broadband light source (NKT Photonics SuperK). The transmitted light was guided to an optical spectrum analyzer via a 1m-long multimode silica fiber. The recorded absorption values at 1.95 μm and 2.22 μm match well to data reported in earlier works [45]. The absorption of CS₂ at shorter wavelengths is very low and thus could not be retrieved in our experiments due to the limited dynamic range of the spectrometer (see Fig. 4c). Therefore, we approximated the absorption below 1.85 μm by a linear dependence of α_m on λ . This extrapolation is important for the consistency of the numerical simulations (only continuous functions are used), but has no influence on the simulation results as the overall modal attenuation for wavelength shorter than 1.8 μm is very low.

R1.2 Similarly, Fig. 5b is not called out in the text and the discussion relating to 5c comes after Fig. 6. This makes it very difficult to understand these simulations when you first encounter them. Also, although I can see slight differences in these two spectrograms, it is not clear why the hybrid case is considered better than the GNLSE. The authors need to better explain why the hybrid Schrödinger equation is preferred over the GNLSE, else it is not clear why you would need to bother with this.

The recent version of the manuscript now calls out Fig. 5b when describing the simulation results. We are aware that the references to Fig. 6 and to 5c are not in strict numerical order. However, we discussed this issue internally in great detail and we wish to leave this order for the following reason: After presenting the experimental data we want to show that all indicators for the new soliton dynamics (Fig. 6) are in accordance with the most generic and widely accepted simulation approach, which is the GNLSE (Fig. 5b). This is of essential importance and forms the basis for our argumentation of the emergence of a new type of temporal soliton dynamic. As a next step (after Fig. 6), we introduce the hybrid nonlinear Schrödinger Equation, which is a simplified version of the GNLSE and directly results from the hybrid nature of the response. From our point of view, this novel type of equation opens up a completely new perspective on how to understand hybrid soliton dynamics on the basis of a straightforward numerical method. The good match between experiment and simulations confirms the validity of this tool, which is therefore very helpful in analyzing each step of the non-instantaneous supercontinuum process, and ultimately allows us to explain hybrid soliton dynamics in terms of soliton trapping in a quasi-static potential. As a result, we strongly believe that the hybrid nonlinear Schrödinger Equation enables new insights into non-instantaneous nonlinear interactions in the future. For these reasons we would like to keep the discussion as it is now, and not separate it from Fig. 5 (which would require, for example, introducing a new Fig. 7), as otherwise a direct comparison of experimental spectra with simulations relying on the GNLSE, and with those related to the hybrid equation, would be exceedingly difficult. To improve the clarity with regards to this point we have added the following sentence at the beginning of the numeric result section (II. 286):

We numerically investigate the supercontinuum process with two types of nonlinear pulse propagation equation, namely a generalized and a hybrid form of the nonlinear Schrödinger equation. Due to its novelty, the latter is discussed with the related results (Fig. 5c) at the end of this section.

However, we agree with the Reviewer that the manuscript undervalues the importance of this new nonlinear Schrödinger Equation. To address this we have extended the discussion section in order to highlight the impact of the hybrid nonlinear Schrödinger Equation (ll. 361):

Although the CS₂/silica system presented here is only one example, we are convinced that the hybrid nonlinear Schrödinger Equation is also applicable to other highly non-instantaneous waveguide systems, as it may form a strong link between the non-instantaneous solitons theoretically predicted by Conti et al. and the states which are observable in realistic hybrid-nonlinear systems. The fact that the hybrid nonlinear Schrödinger Equation describes the observed soliton dynamics very well suggests that the non-instantaneous nonlinear phase plays a major role already in the soliton fission process. In future studies, this Schrödinger equation might help to answer fundamental questions, such as whether the new type of solitary wave actually appears immediately during fission or whether the hybrid nature of those states is imposed on classical solitons during propagation after the actual fission process.

We greatly appreciate that the Reviewer has raised this point.

R2.3 A minor point, the authors use the abbreviation GNSE for the generalized nonlinear Schrodinger equation. GNLSE is more common.

We agree with the Reviewer and have changed it to GNLSE throughout the recent version of the manuscript.

R2.4 Although the text is perfectly understandable, there are still several typos, missing articles, plurals that should be singular etc. Also, when they refer to the pulses being hundreds of femtoseconds – they should mention they are talking about durations here. I would recommend they ask a native English speaker to proofread this, or ask one of the highly experienced co-authors to review it properly.

We apologize for the insufficient language quality. All co-authors carefully read the manuscript again and gave detailed feedback, which was, however, mainly related to the content. The recent version was also spell checked by a scientific language editing service to improve the readability. With that we hope that the English and semantic quality of the manuscript is now to the Reviewer's satisfaction.

Reviewer 2

This paper is greatly improved from the first version. The novelty of the research is much easier to understand (coherent soliton behavior). Importantly, the distinction between well-known solitons and the current work is now much clearer. This also goes for the recent theory work by Conti and its relative degree of applicability to the present case. The shift away from the discussion of 'linearons' and related was also echoed by the other reviewer. The more general 'non-instantaneous nonlinearities' is appreciated.

The revised presentation order is definitely helpful as it first lays out the theory and a physical description of the expected phenomenon before moving onto experiments. Apologies if the first review appeared harsh. While the high technical quality was noted, as written, I feared that the work would be lost on the vast majority of readers. I appreciate the authors took the advice of the reviewers into consideration to improve their work. It appears that the other reviewer suggested similar updates. This work represents a nice piece of science and the authors should keep up the good work.

We greatly appreciate the kind words of the Reviewer and we are glad that we could satisfy each of her/his previous comments on the manuscript. We agree that the manuscript is now greatly improved.

There remain a few points to address before the work is ready for the public eye.

TECHNICAL COMMENTS

Coherence of the system - two major points here.

R2.1 Unity coherence?

Is this indeed fully coherent in Figure 3? While it is obvious that this system is more coherent than regular soliton supercontinuum, $g(1) = 100\%$ would be remarkable and is very surprising. Indeed, one could say it is too good to be true. If this is true, what are the conditions (molecular fraction, dispersion, other) to achieve this? There must be some practical limitations to the system. See the next point.

We like to address this comment in the subsequent response (R2.2).

R2.2 Last version versus this version.

This is all the more shocking given the striking difference between the earlier paper and this one. Figure 3a(left) replaces old Fig. 6a (right). The coherence of these two is completely different. Is all of this simply due to the change of 81 to 85% molecular fraction?

It is recommended the authors outline the sensitivity of this claim especially given the strong claim of full coherence. Thank you for pointing out the 'stiffen' mechanism for the phase noise in your paper.

We thank the Reviewer for raising this issue. We were also pleasantly surprised by the results of the new simulations, showing a further improvement in coherence. We had a very detailed and careful look on the new findings and found that the main reason relates back to the small increase of molecular fraction factor as correctly suggested by the Reviewer. Especially values around $f_m = 80\%$ seem to be critical for the system coherence (and the soliton dynamic in general) at the selected pulse parameters, as indicated in Supplementary Figure 3. Spectral side lobes at the fission point – which indicate noise-triggered modulation instabilities - dominate the generation process at small values of f_m . However, they completely disappear when the molecular fraction factor increases from 70% to 80%. A detailed study of this transition point will be the focus of future work. So far, we can only confirm that even small changes around 80% molecular fraction appear to have a drastic impact on the supercontinuum dynamics (see Fig. R1) for this particular set of parameters (i.e., input power, fiber length, pulse duration). This also emphasizes the important role of an accurate quantitative model of the nonlinear response, which was first published by Reichert et al [Optica 1(6), 436, 2014],

and in particular the erratum to this paper resulted in a significant improvement of the coherence in the revised version of the manuscript, as shown in Fig. R1.

Fig. R1 Evolution of the first order coherence along a CS₂/silica fiber (core diameter: 4.7 μm, sech² pulse shape with 450 fs duration and 1 nJ energy) assuming molecular fractions of (a) 80% (90 individual spectra) and (b) 85% (90 spectra). The linear colormap goes from 0 (black, no coherence) to 1 (white, perfect coherence).

While addressing the Reviewer’s comment above we came across a small mistake in our normalization of the noise model, which has now been fixed. The messages stated in the manuscript remain the same but for a different set of parameters. Further details can be found in the section “Corrections of noise model and the coherence calculations” at the end of this letter. We would like to thank the Reviewer for her/his comment, which motivated us to cross check the noise model used.

We would like to re-emphasize that the scope of this manuscript is to reveal the potential of hybrid liquid-core fiber systems for novel supercontinuum generation regimes, aiming to present first insights into a fundamentally new type of coherence dynamic. In addition to applications requiring a high degree of coherence, our discussion might be essential in understanding other effects, for example the reduced gain of parametric four-wave mixing in liquid-core fibers reported earlier by Barbier et al. [New J. Phys. 17, 053031, 2015]. The low susceptibility to photon (seed) noise in such a system might be a reason for the unexpectedly low FWM gain which they determined to be one to two orders of magnitude lower than in comparable glass fiber systems. Our next goal for a future study is to present a complete picture of the impact of all the different dependencies and parameters on coherence, which will hopefully allow us to define experimental conditions and fiber geometries to obtain optimal coherence for maximizing the spectral bandwidth at high spectral power densities.

R2.3 Small points

- Page 4: The new figure 1 is helpful to understand your system. Thank you.
Thank you, we appreciate this comment.
- Figure 5: What do the (i), (ii), (iii) correspond to? Please define in the text or caption.
We apologize for the confusion at this point. We have added a comment in the figure caption mentioning that the labels correspond to the three different cases discussed in the main text ((i): instantaneous response, (ii) noninstantaneous response, (iii) hybrid response). We have changed the text accordingly.

- Nonlinear loss: It would be helpful to reference the analysis on nonlinear loss in the supplement in main paper, so it's clear to others this is considered.
We would like to kindly point out that we already explained in the Materials & Methods section "Impact of model uncertainties" (now ll. 492) that nonlinear losses can be neglected. However, to address the Reviewer's issue, we added the following sentences to the Materials & Methods section "Supercontinuum measurements and data processing" in the recent version of the manuscript (ll. 428):
The power at the input and output of the fiber was measured with a thermal powermeter before and after each reading (see Suppl. Fig. 6) confirming that nonlinear absorption plays a minor role in our investigation.
- Also note Supplement Fig 6, the legend should have a line instead of dots for the theory.
We appreciate again the thorough analysis of Reviewer. A close check confirmed that what appears as a line in the plot is in fact a series of densely located simulation points, so that the figure legend correctly states "point". We hope that with that we have satisfied the Reviewer's issue.
- RI? This is not defined, but we can guess what it is. Do we really need to abbreviate refractive index?
The Reviewer is correct, thank you. As a residue from a previous version we forgot to explicitly define this abbreviation. In the recent manuscript we instead use the complete term "refractive index" instead of RI, i.e., the phrase "refractive index" is not abbreviated anymore throughout the entire manuscript.
- Definition of gamma: The standard nonlinear parameter is: $\gamma = k_0 n_2 / A_{\text{eff}}$. In the Methods section, the authors have a $A_{\text{eff}}(1/4)$. Is this a different gamma? The units don't work out at present. This section needs to be re-visited.
In fact, we used another definition of γ in the previous manuscript to account for the mode area overlap correction introduced by Laegsgaard (Opt. Exp. **15**(24), 16110, 2007). We understand that this is misleading, in particular since we plot the conventional nonlinear parameter in Fig. 4c and not the modified version. In the recent manuscript, we corrected our definitions. We define gamma in the conventional way, but we introduce a second modified gamma (gamma tilde) which appears in the nonlinear Schrödinger equations to account for the mode overlap correction. We extended the explanation in the Materials & Methods section "Liquid-core fiber dispersion design" (ll. 393):
To account for the dispersion of the mode field area [43] we isolate the factor $A_{\text{eff}}^{-3/4}$ from the conventional definition of γ , which serves as normalization factor of the field amplitudes in the nonlinear term of the Schrödinger Equation (e.g. see Eq. (6)).
The new nomenclature makes the normalization process much more transparent and we greatly appreciate the Reviewer's accurate eye. Thank you.
- Reply letter, page 20: "Higher-order soliton compression is a result of self-steepening."
Higher-order soliton compression results simply from higher-order solitons and does not require self-steepening to occur.
The Reviewer is indeed correct and we appreciate her/his comment, which represents an inaccuracy in the argumentation in the respective sentence in the last response letter. We apologize for this mistake.

GENERAL GRAMMAR CHECK

This was mentioned in the first review. It appears these comments were not taken seriously. There are easily well over ten misspellings, missing words, and related grammar faults in the current version. I do not list them here as that's not the role of the reviewer. While the content comes through, these are a distraction to the reader and could result in them losing their focus. Beyond a careful look themselves, the authors should have some people outside their group read the paper.

In fact, we took her/his comment quite seriously in the first revision. The text was again read and corrected multiple times by all co-authors. In order to fully address the Reviewer's request we have also sent the manuscript to a professional language editing service. We apologize again and hope the quality of the language in the current version of the manuscript is acceptable for the Reviewer.

Corrections of noise model and the coherence calculations

During the revision process, in particular while preparing the response for R2.2, we cross-checked the coherence calculations by comparing our results with published data and found that the input noise background was incorrectly scaled. As a consequence, the input noise of the simulations shown in Fig. 3 was underestimated, which motivated us to recalculate all results related to coherence. To summarize, we obtain the same result – namely that noninstantaneous systems significantly improves the coherence at high soliton numbers - but for a different parameter set. Here we take the opportunity to outline these changes in detail.

A careful check of the implemented one-photon-per-mode noise model revealed an incorrect normalization of the noise amplitude which was an artifact of an earlier version of our code. We corrected our noise model and contacted two key experts in the field of nonlinear light generation, namely Dr. Wonkeun Chang¹ and Prof. John Dudley², with Prof. Dudley being one of the world's primary and most recognized experts in the field of supercontinuum generation. Both experts confirmed the correctness of our revised simulation model, which was checked by comparing test simulations with results obtained by their codes. For example, we compare in Fig. R2 our simulation results now including the corrected noise model with the data published by Dudley *et al* [Rev. Mod. Phys. 78(4), 1135, 2006]. The spectral and coherence evolution match very well. The slightly smaller red-shift of the first fundamental soliton towards the near-infrared is due to the use of a slightly different Raman model. We included Prof. Dudley and Dr. Chang in our acknowledgements as an appreciation of their support.

¹Australian National University, Canberra, Australia (a former colleague of the last author)

²CNRS-University of Franche-Comté, Besançon, France

Fig. R2 Evolution of spectrum and first order coherence of 10 cm silica PCF (parameters defined in the reference) pumped with a 0.6 nJ sech^2 pulse at 800 nm with 100 fs (a) as published in [Dudley et al, Rev. Mod. Phys. 78(4), 1135, 2006], and (b) as calculated with our corrected noise model. Reprinted figure with permission from Dudley, J, Genty, G, and Coen, S. Rev. Mod. Phys. 78, 1135 (2006). Copyright 2006 by the American Physical Society.

Due to the increased noise floor level we had to adapt our input power to obtain a similar message with regard to Fig. 3. The new version of the manuscript contains an updated Fig. 3 and new parameter combinations. It is important to note that the key messages of Fig. 3 remain as before and are as follows:

- a) The first order coherence of hybrid systems is close to unity at conditions instantaneous systems are strongly dominated by modulations instabilities, with the latter suppressing coherence related spectral features.
- b) This high degree of coherence of the hybrid system appears at high soliton numbers $N = 41$, which are significantly higher than the coherence criterion given by Dudley *et al* (i.e. $N < 10$), showing that the hybrid systems clearly outperform the instantaneous system in terms of coherence. For instance, hybrid systems are able to deliver coherent supercontinua at 20 times higher input peak powers compared to instantaneous systems, which is of great importance for many applications.
- c) The spectral distribution of the signatures in the spectra of the hybrid systems remains highly distinguishable from that of instantaneous systems under similar simulation conditions (e.g., same input power and temporal pulse width).

Thus, the corrected model only leads to a shift of the parameter set, with the qualitative argumentation of impact and performance of the non-instantaneous response remaining unchanged (as indicated by the updated Suppl. Fig. 3, too).

Using the corrected noise model we also recalculated the spectrum in Fig. 6a, which shows a comparison of experiment and simulations (average of 100 individual runs with input noise). However, the increased noise level in simulations leads to flatter average spectra (Fig. R3b), which resembles the experimentally measured spectra less (e.g., the 10 dB spectral modulations at high input powers are not clearly visible in simulations). After many calculations and discussions with the Prof. Dudley and Dr. Chang we are convinced that in our experimental situation, the one-photon-per-mode noise model is not the only source of noise, i.e., the current measurements might include noise sources with different frequency dependence (e.g., noise from the mode locked fiber pump laser, Raman noise) than in case of white noise. If those noise frequency components do not match with the phase matching condition of modulation instabilities, other nonlinear processes are more

likely to dominate. According to Prof. Dudley, the one-photon-per-mode noise model is in fact an empirical model without deep physical justification, which has led to surprisingly good results in the case of modulations instabilities in glass fibers. Nevertheless, a verification of the one-photon-per-mode model was not possible so far since instantaneous systems are obviously highly susceptible to any kind of input noise.

The distinction between the noise models might only play a minor role in instantaneous systems due to their higher susceptibility to any level of noise in the proximity of the phase matching frequencies of the modulations instabilities, as for example discussed in the response to R2.2. However, in the phase-stabilizing non-instantaneous systems this distinction might become observable. As a further example, the overwhelming susceptibility to noise of instantaneous systems was demonstrated in gas-filled hollow-core fibers pumped with 500 fs pump at soliton numbers as large as 90 and higher [Phys. Rev. Lett. **111**, 033902, 2013]). The authors of this work observe different spectral features compared to our observations at soliton numbers being of the same order as in our experiment (e.g., they observe modulation instability driven side lobes at the fission point and smeared out spectra for high input powers). This again confirms the fundamental differences in the broadening dynamics between an instantaneous and non-instantaneous system at similar pump conditions.

As a result of these new insights and the lack of knowledge about other types of noises in our experimental configuration, we have replaced the simulation spectrum in Fig. 6a by a single shot spectrum which contains no input noise (see Fig. R3a) and resembles the experimentally measured spectra sufficiently well. We also changed the description of Fig. 6a in the text (ll. 314) to:

The modulation contrast of the fringes on the soliton side is of the order of 5 to 10 dB and matches sufficiently well to noise-free simulations (Fig. 6a).

Fig. R3 Comparison between measured and simulated spectra after 7 cm $\text{CS}_2/\text{silica}$ fiber (ID 4.7 μm) pumped with 460 fs and 7 nJ ($N = 108$) using (a) a single spectrum without input noise and (b) an average spectrum of 50 individual runs with the one-photon-per-mode noise model.

In summary the correction of our model did not affect the core messages of the manuscript, which is to present four measurable indicators for new hybrid soliton dynamics (i.e., dispersive wave phase matching, reduced spectral bandwidth, higher fission energy, missing features of modulations instabilities). We are glad to have found this mistake at this early stage, showing that the peer-review process is invaluable in ensuring the high quality of scientific publications.

REVIEWERS' COMMENTS:

Reviewer #2 (Remarks to the Author):

I read the updated manuscript and letter. This manuscript is much improved since the original submission.

It's always a challenge to convince our peers of 'new' science and certainly the lengthy review process has benefited both the authors and the reviewers.

Given that many people in the field would never read anything on this topic, there is one last point that would help attract a broader audience to the novelty of the findings in this paper. Fortunately, the authors have already done the hard work and this is a superficial modification.

\Visual demonstration of coherence

Given the sensitivity of the system coherence to molecular fraction (81% vs. 85%), the authors should add a line in the main text highlighting this almost Heaviside response in a very direct and obvious way. This is obviously a fundamental property of the system that the authors should highlight both with a few lines of text, and by placing Figure R1 in the main text.

Figure R1 is a spectacular visual explanation of this effect. It looks like it would fit between figures 3A and 3B) as it is a fundamental piece of information for this physical system.

\Thank you for your effort and dedication to seeing this work through.

Thank you for providing further feedback to our revised manuscript. We are grateful to the Reviewers for the positive feedback and for the important points which have helped us to correct and improve the manuscript. In this letter, we respond on each comment of the two Reviewers and on the formatting issues. Please find our answers in blue and the changes in the recent manuscript in *italic red*.

Reviewer 2

I read the updated manuscript and letter. This manuscript is much improved since the original submission.

It's always a challenge to convince our peers of 'new' science and certainly the lengthy review process has benefited both the authors and the reviewers.

Given that many people in the field would never read anything on this topic, there is one last point that would help attract a broader audience to the novelty of the findings in this paper. Fortunately, the authors have already done the hard work and this is a superficial modification.

We thank the Reviewer for her/his decision and highly appreciate her/his thoughtful input along the entire review process. We are happy to apply the proposed changes to increase the impact of the manuscript.

R2.1 Visual demonstration of coherence

Given the sensitivity of the system coherence to molecular fraction (81% vs. 85%), the authors should add a line in the main text highlighting this almost Heaviside response in a very direct and obvious way. This is obviously a fundamental property of the system that the authors should highlight both with a few lines of text, and by placing Figure R1 in the main text.

Figure R1 is a spectacular visual explanation of this effect. It looks like it would fit between figures 3A and 3B) as it is a fundamental piece of information for this physical system.

We agree with the Reviewer that the sudden decay of the coherence for a minimal reduction of the molecular fraction is important information for the reader. We actually tried to conduct the suggested change (i.e., inclusion of Fig. R1 into Fig. 3) but, however, found that there is not sufficient space to include the results directly into Fig. 3, as otherwise this figure would give an overloaded

impression. Another idea was to include the coherence for case of $f_m = 0.8$ as a grey line in Fig. 3a, which, however, also clutters up the figure and distracts the reader from the main storyline. Therefore, we prefer to place the color plots for the two cases (i.e., former Fig. R1) into the supplementary information.

Fig. R1: Remake of Fig. 3 (not used in the main manuscript) from the main manuscript with additional coherence spectrum in panel (a) for the case of 80 % molecular contribution.

To highlight the drastic dependence of the coherence on the molecular fraction in the manuscript we added the following two sentences close to Fig. 3, clearly pointing out this behavior and referring to Suppl. Fig. 4 for the interested reader:

Note that the high level of coherence critically depends on the actual value of the molecular fraction. For the system parameters chosen in this work, only 5 % less molecular contribution (i.e., $f_m = 0.8$) reduces the first-order coherence significantly (see Supplementary Fig. 4).

Finally, we want to inform the Reviewer that we changed the model fit parameters in Tab. 1 of the Methods section in the manuscript. We found that terms higher than β_5 do not contribute significantly to the fiber dispersion anymore, so we took out the column for β_6 . The recent fit parameters approximate the theoretical fiber dispersion much better now. This change has no influence on any result in the manuscript or the supplementary information, since it is only a support for the curious reader.

We thank the Reviewer once again for her/his thoroughness and patience throughout the many review steps of our manuscript. We think the manuscript significantly gained quality through this entire process.